# A local-ingredients-based supplement is an alternative to corn-soy blends plus for treating moderate acute malnutrition among children aged 6 to 59 months: A randomized controlled non-inferiority trial in Wolaita, Southern Ethiopia

**Debritu Nane**[1,2,3]*, **Anne Hatløy**[2,4], **Bernt Lindtjørn**[1,2]

**1** School of Public and Environmental Health, Hawassa University, Awassa, Ethiopia, **2** Centre for International Health, University of Bergen, Bergen, Norway, **3** Wolaita Sodo University, Wolaita Sodo, Ethiopia, **4** Fafo Institute for Labour and Social Research, Oslo, Norway

* nanedebritu@gmail.com

## Abstract

### Background

Globally, moderate acute malnutrition (MAM) affects approximately 5% of children below five years of age. MAM is a persistent public health problem in Ethiopia. The current approach in Ethiopia for managing MAM is a supplementary feeding program; however, this is only provided to chronically food-insecure areas. The objective of the study was to compare a local-ingredients-based supplement (LIBS) with the standard corn-soy blend plus (CSB+) in treating MAM among children aged 6 to 59 months to test the hypothesis that the recovery rate achieved with LIBS will not be more than 7% worse than that achieved with CSB+.

### Methods and findings

We used an individual randomized controlled non-inferiority trial design with two arms, involving 324 children with MAM aged 6 to 59 months in Wolaita, Southern Ethiopia. One hundred and sixty-two children were randomly assigned to each of the two arms. In the first arm, 125.2 g of LIBS with 8 ml of refined deodorized and cholesterol-free sunflower oil/day was provided. In the second arm, 150 g of CSB+ with 16 ml of refined deodorized and cholesterol-free sunflower oil/day was provided. Each child was provided with a daily ration of either LIBS or CSB+ for 12 weeks. Both intention-to-treat (ITT) and per-protocol (PP) analyses were done. ITT and PP analyses showed non-inferiority of LIBS compared with CSB+ for recovery rate [ITT risk difference = 4.9% (95% CI: -4.70, 14.50); PP risk difference = 3.7% (95% CI: −5.91, 13.31)]; average weight gain [ITT risk difference = 0.10 g (95% CI: -0.33 g, 0.53 g); PP risk difference = 0.04 g (95% CI: -0.38 g, 0.47 g)]; and recovery time [ITT risk difference = -2.64 days (95% CI: -8.40 days, 3.13 days); PP difference -2.17 days

**Data Availability Statement:** All relevant data are within the paper and its Supporting Information files.

**Funding:** The author who received the award of this study is Debritu Nane (corresponding author). The grant number of the award is ETH-13/0025. The funding organization of this research is NORAD (Norwegian Agency for Development Cooperation); through the NORHED program (Norwegian Program for Capacity Development in Higher Education and Research for Development). URL of the funder is "South Ethiopia Network of Universities in Public Health (SENUPH) improving women's participation in post graduate education". This funding organization is responsible in the process of designing the study and data collection, analysis, and interpretation and in writing the manuscript.

**Competing interests:** The authors have declared that no competing interests exist.

(95% CI: -7.97 days, 3.64 days]. Non-inferiority in MUAC gain and length/height gain was also observed in the LIBS group compared with the CSB+ group.

## Conclusions

LIBS can be used as an alternative to the standard CSB+ for the treatment of MAM. Thus, the potential of scaling up the use of LIBS should be promoted.

## Trial registration

Pan-African Clinical Trial Registration number: PACTR201809662822990.

## Introduction

Globally, acute malnutrition remains a major public health problem, affecting an estimated 55 million children below five years of age [1, 2]. Acute malnutrition occurs when there is severe weight loss [3] possibly caused by inadequate energy and protein intake [4], and infections. Such events frequently occur during periods of prolonged food insecurity, poverty, poor feeding practices, and inadequate household food accessibility [5, 6]. When a repeated episode of acute malnutrition occurs, often along with infections, it can lead to stunting (chronic malnutrition) [3, 7, 8]. Sometimes, especially if the child is less than two years of age, it can also affect other developmental features such as brain impairment [9].

Acute malnutrition is categorized as moderate or severe [7]. Moderate acute malnutrition (MAM) is defined as weight-for-height z-score (WHZ) between -3 and -2 z-scores of the WHO Child Growth Standards median and/or mid-upper arm circumference (MUAC) between ≥11.5 cm and <12.5 cm without bipedal edema [10–12]. Severe acute malnutrition (SAM) is defined as WHZ <-3 SD or a MUAC of <110 mm, or the presence of nutritional edema [7, 13]. According to the Ethiopian Demographic Health Survey (EDHS) 2011, about 10% of children aged below five years of age suffered from acute malnutrition, and among these, 70% had MAM [14]. Children with MAM have a three-fold higher risk of morbidity and mortality than their normal counterparts [1, 15].

There is presently no standardized approach for managing MAM, especially in relatively food secure settings of low- and middle-income countries [12, 16, 17]. In Ethiopia, the current approach for the management of MAM is a supplementary feeding program (SFP), although this is limited to areas where chronic food insecurity exists. In districts not determined to be persistently food insecure, there are no food supplementation strategies for children with MAM. In these districts, the approach to managing children with MAM includes vitamin A supplementation, deworming, and dietary counseling delivered to the caregivers of children [14, 18].

Children with MAM are commonly treated with fortified blended flours, especially corn-soy blend (CSB), super cereal/corn-soy blend plus (CSB+), and super cereal plus/corn-soy blend plus with milk and oil (CSB++) [12, 15, 19]. The current food blends used for managing MAM are CSB+ and CSB++. CSB+ contains corn (64%), whole soya beans (24%), sugar (10%), vegetable oil, and vitamin and mineral premix. CSB++ is made from corn (58%), dehulled soya beans (20%), dried skim milk powder (8%), sugar (10%), vegetable oil, and vitamin and mineral premix; it is used as a complement to breast milk. The combinations of the cereal (corn or maize) and the legume (soya bean) were refined, blended, and precooked (by roasting) for children with MAM below five years of age [12]. These standard food

supplements are effective in treating MAM; however, such management is temporary and unsustainable for managing recurrent malnutrition in poverty-hit countries [15].

The dietary management of MAM could be done by using locally accessible nutrient-dense foods [11]. Local ingredients that are available and accessible can be developed at the household level; they are easy to develop, cheap, and yet contain adequate amounts of the required nutrients [15]. In this paper, the new supplement is referred to as a local-ingredients-based supplement (LIBS); it is made of locally available ingredients such as pumpkin seed, peanut, amaranth grain, flaxseed, and emmer wheat. The proportions of these ingredients were designed to have the required amount of nutrients for managing MAM among children aged 6 to 59 months [11].

Different studies showed that CSB+ had similar effects of treating MAM compared to other food supplements prepared for the treatment of MAM [10, 15, 20]. The differences are too narrow to feasibly conduct a superiority trial. We, therefore, conducted a non-inferiority trial to determine whether LIBS is at least as effective as CSB+ in treating MAM.

To resolve the limited delivery of supplements among children with MAM, the usefulness of LIBS needs to be explored. Besides this, adequate research has not been conducted in Ethiopia on supplementary food that was developed using locally available ingredients, even though it has the potential to contribute to treating MAM. This study aims to evaluate if children aged 6 to 59 months suffering from MAM, treated with 125.2g of LIBS and 8 ml of sunflower oil/day, for 12 weeks, would not have an inferior recovery rate compared to similar children treated with 150 g CSB+ and 16 ml of sunflower oil/day (with a 7% margin of non-inferiority).

## Materials and methods

### Ethics approval and consent to participate

This study was approved by the Hawassa University College of Medicine and Health Sciences Institutional Review Board (IRB/024/10) and Regional Committees for Medical and Health Research Ethics in Norway (2018/69/REK vest). The approval obtained from the institutional review boards covered all sites included in the study. The purpose of the study and methods of data collection, confidentiality, and voluntary participation were explained to the mothers of children who were invited to sign an informed consent form. Verbal (getting thumb marks after reading the information) and written informed consent were obtained from all caregivers of children who met enrollment criteria before the recruitment of their children into the study. All interviews and intervention procedures were conducted in privacy.

This trial was registered at Pan-African Clinical Trial Registration as PACTR201809662822990. The authors confirmed that all ongoing and related trials for these food supplements were registered.

### Subject and setting

We conducted the study in Damot Pulassa district in the Wolaita zone in the South-Western part of Ethiopia, where there is a high level of food insecurity and child malnutrition. Damot Pulassa is characterized by having fragmented farm and land ownership. Damot Pulassa has the highest population density in Ethiopia reaching up to 600 persons per square kilometer in some *Kebeles* (smallest administrative units) of it. The discrepancy between population and land balance has by far continued to be the primary cause of endemic food insecurity in the area [21]. According to a study protocol of this trial, Damot Pulassa district is characterized as maize and root crop livelihood area as these are the main crops cultivated in the district [5]. This district has five health centers and 23 health posts (one per *kebele*) that are led by health extension workers. These health facilities deliver nutrition-linked services like counseling on

feeding practices, screening for nutritional status of young children, and nutritional management of acutely malnourished children.

Damot Pulassa district (Woreda) was purposely selected based on a consideration of the high level of food insecurity, high level of child malnutrition, and access to transportation. All the kebeles in the district had similar admission loads to the management program of MAM, for practical reasons, six out of the 23 kebeles were randomly selected for the study. Children aged 6 to 59 months with MAM, living in selected *kebeles* of Damot Pulassa district were included in the study. Children were excluded if they had SAM (based on WHO 2009 child growth standards), and/or if they had bilateral pitting edema [22], or had any illness or other medical complications that prevented the children from safely consuming supplementary food. The trained research teams and HEWs assessed the children for SAM. Medical complications of children were assessed by the senior pediatric nurse working in the below five years clinic. Children were also excluded if they were simultaneously involved in another supplementary feeding program. Recruited children were identified as MAM according to their MUAC values, and not only according to the values of weight-for-height/length z-scores. The respondents were mothers or caregivers of selected children.

## Study design and intervention

This study was a randomized, controlled, non-inferiority trial that assessed the efficacy of 125.2 g of LIBS with 8 ml of refined deodorized and cholesterol-free sunflower oil/day (the intervention), compared with conventional treatment, which is CSB+ in the amount of 150 g of CSB+/day with 16 ml of refined deodorized and cholesterol free-sunflower oil (the control), in treating MAM for 12 weeks.

## Outcome variables and quality measurements

The primary outcome was recovery rate: percentage of children who attained a MUAC $\geq$12.5 cm and/or WHZ $\geq$ -2 without bipedal edema at the end of 12 weeks. The secondary outcomes were the mean recovery time (the duration within 12 weeks in which the child recovered from MAM) and average weight gain. Children who progressed to SAM during the study or persisted as moderately malnourished at the end of the 12-week follow-up were considered to have failed management for MAM. The selection of outcome measures was based on similar studies [15, 20, 23].

There were two research teams, each team consisted of one supervisor, six data collectors, and eighteen food distributors (selected females living in the study area). The supervisors and data collectors were trained by the principal investigator for seven days before the start of study. The training covered the objectives of the study, data collection systems, questions found in the questionnaire, interview techniques, and anthropometric measurements. Training in anthropometric measurement techniques, periodic standardization, and daily calibration of equipment was done based on WHO recommendations for anthropometric measurement protocols [24]. Before study start, the female food distributors were trained for five days on a daily supplement distribution, cooking of supplementary foods such as porridge and feeding of a child.

When the child was a twin, we provided an additional amount of supplementary food to the caregiver to ensure that the enrolled child was fed with a full portion. When two children with MAM were found in the same household, we provided similar supplements for both children but only the randomly selected child was taken as a study subject. When the recruited child was not at home during the time of follow-up, data collectors and food distributers revisited such households until they found the child. Supervisors oversaw activities daily. The food

distribution process, feeding techniques, and use of the provided food supplements were monitored among randomly selected households on a twice-monthly basis.

## Sample size calculation

We calculated the sample size to investigate if LIBS was not inferior to CSB+ in terms of the recovery rate among children with MAM aged 6 to 59 months. The sample size was calculated based on 80% power of the test, 7% margin of non-inferiority, and assuming a recovery rate with CSB+ of 67%. Considering the above assumptions, a total of 324 children (162 subjects per study group), allowing for a 10% withdrawal rate were required to be sure that the lower limit of a two-sided 95% confidence interval (CI) was above the -7% margin of non-inferiority. The anticipated 10% dropout rate was used based on the observed dropout rate reported in two studies [11, 24]. The non-inferiority margin (-7%) was set depending on the previous studies that showed the comparator group (children who received CSB) had a recovery rate from MAM of 67% [19] and the recovery rate from MAM without any treatment was 54% [18], hence the difference is 13%. The non-inferiority margin was specified considering that the LIBS group has at least a 7% (the average of the difference of two proportions) higher recovery rate than in a group with no supplementary food (placebo). This was based on the recommendations of selecting a non-inferiority margin [25] For this sample size calculation, the Pharma-School sample size calculator for non-inferiority trials was used.

## Recruitment of study participants

The trained data collectors, accompanied by health extension workers, visited all households found in selected *kebeles* (Waribira Golo, Bibiso Olola, Waribira Suke, Shanto, Tomtome, and Lera) in Damot Pulassa district with children aged 6 to 59 months, to assess children for eligibility by measuring MUAC. The recruitment was done from August 27 to September 20, 2018. Children with MUAC <12.5 cm were recorded and brought to the actual screening site with their mother/caregiver, where MUAC was re-measured, and weight and height or length were taken. In addition to the values of MUAC (i.e. between ≥11.5 cm and <12.5 cm), we used the WHZ or weight for length z-score for recruitment (i.e. WHZ: between −3 and −2 Z-scores). When the child was identified as moderately malnourished according to the MUAC and not according to the values of weight-for-height/length z-scores, we recruited them according to their MUAC values. Children aged 6 to 23 months were measured for weight and length, and children aged 24 to 59 months were measured for weight and height. Edematous malnutrition was also assessed using the bilateral pitting edema criterion. The screening for MAM was continued until the sample size was met.

## Randomization

A computer-generated randomization list that contained codes was prepared using random allocation software. The allocation ratio was 1:1. Mothers selected an envelope containing coded numbers that matched with one of the two supplementary foods. A research assistant who was not otherwise involved with the study implemented the randomization process. After randomization, the investigator further classified children into sub-groups, with equal numbers of subjects based on their neighborhood. This was done to facilitate assigning one food distributor per sub-group, where food distributors simply had to access the households with a selected child. The food distribution process was organized by the research assistant who was aware of which number corresponded to which food. This person did not take part in the food distribution.

## Blinding

This study was double-blinded, that are, caregivers, data collectors, and food distributors were blinded for the intervention. Both supplements were packed with the same plastic packs, similar in color and texture, and they were prepared and distributed in the same way. The sugar which was provided for the intervention group was blended with the flour before packing. Sunflower oil was distributed with the supplement for both intervention and control groups. Food distributors were assigned to support caregivers in cooking the porridge with 8 ml of oil for the intervention group and with 16 ml of oil for the CSB+ group. We distributed the oil with colored plastic cups.

## Data collection and follow-up

Before introducing the interviews, the study questionnaires were subjected to pilot-testing and were refined for clarity and correctness. The interviews were conducted with selected caregivers of children with MAM aged 6 to 59 months, for collecting baseline information such as socio-demographic and economic status, child's age, dietary habits, breastfeeding practices, and history of child and maternal illness. The child's weight was measured with a Seca weight scale to the nearest 0.1 kg. The data collectors ensured that the scale was positioned on a flat, firm surface, and weighing was done with light clothing.

The length was measured to the nearest 0.1 cm, using a locally prepared wooden measuring board for children aged 6 to 23 months. For children aged 24 to 59 months, height was measured to the nearest 0.1 cm using a Seca height scale. Before height measurement, the data collectors ensured that the height board was on level ground and the child was barefoot; the collector kneeled to get to the level of the child and encouraged the caregiver to help. For length, data collectors measured the child lying down, being sure that the length board was placed on a flat and stable surface.

MUAC was assessed by non-stretchable standard United Nations Children's Fund (UNICEF) plastic tape measures. The measurement was taken halfway between the acromion and olecranon processes, with the measuring tape fitting comfortably, but without making a depression on the left upper arm. This was done twice for every child, using two different data collectors, and the average of the two measures was recorded to the nearest 0.1 cm. Participants in both groups were visited every week for consecutive 12weeks to collect anthropometric measurements using MUAC, and information on supplement use and morbidity. Every month, anthropometric data were collected with identical equipment as used at baseline. Bilateral pitting edema was assessed by pressing for three seconds on the dorsum of the foot. Based on the follow-up measurement of anthropometry, children who developed SAM were sent to the SAM clinic.

## Description of interventions and distribution of supplements

Subjects in the intervention group received a daily ration of 125.2 g of LIBS with 8 ml of refined deodorized and cholesterol-free sunflower oil for 12 weeks. The composition of LIBS was: 30 g of pumpkin seed, 25 g of peanut grain, 20 g of amaranth grain, 15 g of flaxseed, 10 g of emmer wheat, and 25.2 g of cane sugar with 8 ml of refined deodorized and cholesterol-free sunflower oil. This supplement (one serving) yielded 699 kcal, 22.6 g protein, 56.9 g carbohydrate, and 40.89 g fat. The cane sugar was added to LIBS in which the taste of LIBS made better and the amount of calories that should come from carbohydrates was improved but still lower than the level of carbohydrate found in the CSB+ (conventional food provided for children with MAM in the control group). Likewise, children in the control group received 150 g of CSB+/day with 16 ml of refined deodorized and cholesterol-free sunflower oil; this yielded 751

kcal, 21.25 g protein, 95 g carbohydrate, and 31.76 g fat daily for 12 weeks. The participants from both groups were served the supplements in the morning as breakfast (Table 1). Children aged 6–23 months and 24 to 59 months received a similar amount of food [24]. The food distributors who were trained for the preparation of porridge visited each household daily to provide the supplement and help the caregivers in the preparation of the porridge and feeding, to advise, assess, and resolve problems with feeding. The 150g of CSB+ was diluted with 600 ml water and cooked for 10 to 15 minutes whereas the 125.2g of LIBS was diluted with 400 ml water and cooked for 10 to 15 minutes. The food distributors checked and measured the amount of supplements consumed by the children using the local measuring cup with numbers. In case of not consuming all the prepared porridge, re-feeding was done.

## Data analysis

The field supervisors controlled all data collection sheets for completeness. Data entry was done using Epi Data v. 3.1. (Odense, Denmark). Double data entry was done simultaneously to ensure data quality. According to the recommendations for analyzing and reporting equivalence and non-inferiority trials, both intention-to-treat (ITT) and per-protocol (PP) analyses were done, and the 95% CI level was used to interpret any differences [24]. The ITT analyses involved all patients who were randomly allocated, whereas the PP analysis excluded children who refused, transferred out of the management program, or were lost to follow-ups, but included children who died, or were discharged as cured or non-cured. Statistical analysis was done using SPSS v. 20 (IBM Chicago, IL, U.S.A.) and STATA 15 (StataCorp LLC) software.

Comparisons of baseline and outcomes characteristics between CSB+ and LIBS were summarized as percentages and means (±SDs). Comparison of outcomes between the LIBS and CSB+ groups were made by using a chi-square test for categorical variables and generalized estimating equations (GEE) for continuous variables. Differences in the estimated proportion

**Table 1. Nutrient composition of the supplementary foods.**

| Nutrient | 125 gm of LIBS with 8 ml sunflower oil | 150 gm of CSB+ with 16ml of sunflower oil |
|---|---|---|
| **Energy (kcl)** | 698.5 | 750.84 |
| **Protein (g)** | 22.6 | 21.25 |
| **Carbohydrate (g)** | 56.9 | 95.0 |
| **Fat (g)** | 40.89 | 3176 |
| **Ash (g)** | 2.1 | 4.3 |
| **Iron (mg)** | 8.1 | 6.00 |
| **Zn (mg)** | 5.6 | 7.5 |
| **Calcium (mg)** | 100.00 | 195.00 |
| **Phosphorous (mg)** | 470.55 | 300.00 |
| **Potassium (mg)** | 666.14 | 600.00 |
| **Magnesium (mg)** | 394.7 | 107.75 |
| **Sodium (mg)** | 84.6 | 41.25 |
| **Folic acid (μg)** | 49.4 | 90.00 |

Abbreviations: LIBS: local-ingredients-based supplement; CSB+: corn-soy blend plus; kcal: kilocalorie; g: gram; and mg: milligram.

**Note:** Nutrient values for the LIBS ration were calculated by using the United States Department of Agriculture (USDA) food composition database and NutriSurvey software. Nutrient values for the CSB+ were adapted from Amegovu KA, Ogwok P, Ochola S, Yiga P, Musalima HJ, Mutenyo E. Formulation of sorghum-peanut blend using linear programming for treatment of moderate acute malnutrition in Uganda. J Food Chem and Nutr. 2013; 1(2):67–77.

between the groups along with 95% CI were estimated to infer non-inferiority. Our trial is testing against a one-sided hypothesis, but the decision of non-inferiority is based on a two-sided 95% confidence interval, even if we deal with the lower limit of the interval. The recovery time between groups was predicted in Kaplan-Meier (log-rank) curves of survival analysis. Mean difference in weight gain, mean difference of MUAC gain, and mean difference in height/length gain were computed to describe the magnitude of the difference between the two groups. Rates of weight gain through the whole period of follow-up were estimated in g/kg body weight/day and compared between the study groups. These were calculated by dividing the weight gain (weight at exit minus weight at admission), expressed in grams, by the weight at admission (in kilograms), and the length of stay (in days). Anthropometric indexes were calculated using ENA for SMART 2011 software.

## Results

### The enrollment process and trial profile for the study participants

A total of 1006 children were screened for MAM, of whom 682 children were excluded because 650 children were without MAM and 32 children did not meet inclusion criteria. In the study, 324 children with MAM aged 6 to 59 months were enrolled to either LIBS ($n = 162$) or CSB+ ($n = 162$). Of these, 311 (96%) children with MAM and their mothers/caregivers completed the study, with seven lost to follow-up and six discontinued due to SAM. The reason for all those lost to follow-ups was moving from the study site. The enrollment and allocation of participants, lost to follow-ups, discontinued, and completed the study are illustrated in (**Fig 1**). No serious adverse reactions were detected.

### Baseline characteristics of the study participants

Baseline characteristics were compared between the LIBS and CSB+ groups (Table 2). The mean (SD) age of the mothers in the LIBS group was 31.5 (5.91) years and 32.27 (6.27) years in the CSB+ group. The mean (SD) age of the children in the LIBS group was 21.96 (11.32) months and 21.18 (11.81) months in the CSB+ group. Most children were from agricultural households, with six to eight persons per household and with a low rate of literacy. More than half of children, 54.6% of children in the LIBS group, and 64.2% of children in the CSB+ group had morbidity history based on two weeks recall.

### Outcomes in the two intervention arms over 12 weeks

Fisher's exact test showed that the overall proportion of children who recovered was 72.2%, whereas 1.9% deteriorated to SAM, 24.4% remained in MAM, and 1.5% was lost to follow-ups at the end of the 12-week intervention.

The proportion of children who recovered was similar for LIBS and CSB+ groups in both ITT and PP analysis. ITT analysis showed recovery of 75.90% for LIBS (95% CI: 68.76, 81.89) and 71.0% for CSB+ (95% CI: 63.56, 77.44) (p = 0.314), whereas the PP analysis showed recovery of 76.58% for LIBS (95% CI: 69.37., 82.54) and 72.90% for CSB+ (95% CI: 65.39, 79.30) (p = 0.454) The risk differences for LIBS compared with CSB+ were 4.9% (95% CI: -4.70, 14.50) in ITT analysis and risk difference = 3.7% (95% CI: -5.91, 13.31) in PP analysis (Table 3). Both PP and ITT analyses showed that LIBS was non-inferior compared with CSB + in terms of the recovery rate, as the entire CI of the difference between the two groups was above the predefined non-inferiority margin of risk difference (-7%). Besides, both PP and ITT showed that LIBS is not significantly different from CSB+ because the two-sided 95% CI crosses the 0 outcome difference (**Fig 2**).

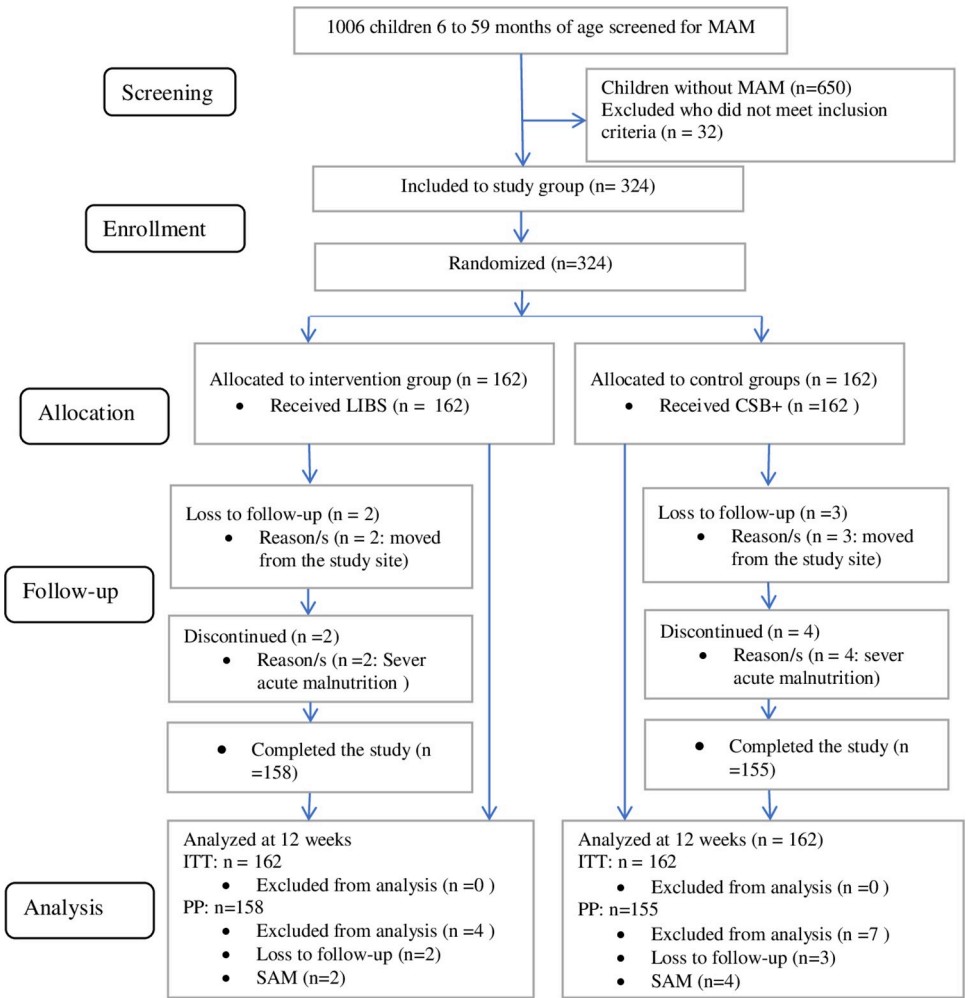

**Fig 1. Flowchart of study enrollment to completion.** Screened for MAM, children aged 6 to 59 months who screened for moderate acute malnutrition; Children without MAM, children aged 6 to 59 months who did not have moderate acute malnutrition; Randomized children aged 6 to 59 months who were assessed positive for moderate acute malnutrition, fulfilled the inclusion criteria, were recruited and randomly allocated; The loss to follow-up, children who were randomly allocated to intervention and control groups and stopped participating in the study at any stage of the study; Discontinued due to SAM, children who were randomly allocated to intervention and control groups and discontinued from the study at any stage of the study; Analyzed at 12 weeks, children aged 6 to 59 months whose information was analyzed at 12 weeks; ITT and PP analysis are done. Abbreviations: MAM: moderate acute malnutrition; LIBS: local-ingredients-based supplement; CSB+: corn-soy blend plus; SAM: severe acute malnutrition; ITT: intention to treat; PP: per-protocol.

The daily weight gain was similar in both groups. In ITT analysis, the mean weight gain (SD) among children in the LIBS group was 3.18 (1.89) g/kg/day, whereas the mean weight gain (SD) among children in the CSB+ group was 3.09 (2.05) g/kg/day (p = 0.642) The PP analysis showed that the mean weight gain (SD) among children in the LIBS group was 3.25 (1.86) g/kg/day, whereas the mean weight gain (SD) among children in the CSB+ group was 3.21 (1.98) g/kg/day (p = 0.838). Both ITT and PP analyses showed that LIBS group was non-inferior compared with the CSB+ group (predefined non-inferiority margin of risk difference = -1.3 g/kg/d) [ITT risk difference = 0.10 (95% CI: -0.33, 0.53); PP risk difference = 0.04 (95% CI: -0.38, 0. 47)] (**Fig 3 and Table 3**). The ITT analysis showed that the mean MUAC gain (SD) for the LIBS group was 0.12 (0.07) mm/day and 0.11 (0.08) mm/day for the CSB+ group.

**Table 2. Baseline characteristics of study subjects per treatment and control group.**

| Characteristics (N = 324) | LIBS (n = 162) | CSB+ (n = 162) |
|---|---|---|
| **Sex of child [n (%)]** | | |
| Male | 67 (41.4) | 70 (43.2) |
| Female | 95 (58.6) | 92 (56.8) |
| **Age of the child in months [mean (SD)]** | 21.96 (11.32) | 21.18 (11.81) |
| **Age of the mother in years [mean (SD)]** | 31.5 (5.91) | 32.27 (6.27) |
| **Maternal MUAC (cm), [mean (SD)]** | 22.01(1.48) | 22.17 (1.55) |
| **The education level of respondents [n (%)]** | | |
| Non-formal education | 111 (68.5) | 112 (69.1) |
| Primary | 46 (28.4) | 37 (22.8) |
| Secondary and above | 5 (3.1) | 13 (8.1) |
| **Occupation of the household head [n (%)]** | | |
| Farmer | 95 (58.6) | 73 (45.1) |
| Daily worker | 25 (15.4) | 41 (24.3) |
| Business owner | 30 (18.6) | 33 (20.4) |
| Monthly paid worker | 6 (3.7) | 8 (4.9) |
| Handcrafts | 6 (3.7) | 7 (4.3) |
| **Morbidity based on two weeks recall [n (%)]** | | |
| Yes | 90 (55.6) | 104 (64.2) |
| No | 72 (44.4) | 58 (35.8) |
| **Household size [n (%)]** | | |
| 3 to 5 | 42 (25.9) | 47 (29.0) |
| 6 to 8 | 96 (59.3) | 102 (63.0) |
| >8 | 24 (14.8) | 13 (8.0) |
| **Use of bed net by the child [n (%)]** | | |
| Yes | 128 (79.02) | 116 (71.6) |
| No | 34 (20.98) | 46 (28.4) |
| **MUAC of the child (cm), mean (SD)** | 12.0 (0.36) | 12.05 (0.32) |
| **Weight of the child (kg), mean (SD)** | 8.03 (1.23) | 7.94 (1.26) |
| **Height of the child (cm), mean (SD)** | 76.33 (6.68) | 76.29 (6.79) |
| **WHZ, mean (SD)** | -2.29 (0.47) | -2.33 (0.29) |

LIBS: local-ingredients-based supplement; CSB+: improved corn-soy blend; WHZ: weight-for-age Z score; MUAC: mid-upper arm circumference.

In PP analysis, the mean MUAC gain (SD) for the LIBS group was 0.12 (0.07) mm/day and 0.11 (0.07) mm/day for the CSB+ group. Both ITT and PP analyses showed that there is no difference in MUAC gain between the two groups (p>0.05). There was no difference in daily length/height gain over 12 weeks between the LIBS and CSB+ groups (p>0.05) (Table 3).

Overall, the mean (SD) recovery time (predefined non-inferiority margin of risk difference = 14 days) in the ITT analysis for the LIBS group was 54.27 days (26.74) and for the CSB + group, it was 56.9 days (25.99), giving a difference of -2.64 days (95% CI: -8.40, 3.13 days). In PP analysis, the mean recovery time (SD) for the LIBS group was 53.52 days (26.65) and for the CSB+ group, 55.68 days (25.91), giving a difference of -2.17 (95% CI: -7.97, 3.64 days) (Table 3). Both ITT and PP analysis showed that the mean recovery time in the LIBS group is non-inferior to the CSB+ group (**Fig 4**).

There is no difference in recovery time between the LIBS and CSB+ groups in ITT analysis p = 0.368 (**Fig 5**) and in PP analysis, p = 0.466 (**Fig 6**).

**Table 3. Outcomes in the two intervention arms over 12 weeks.**

| Outcomes | All (N = 324) | LIBS (n = 162) | CSB+ (n = 162) | Proportion/mean difference with its 95% CI | P-value |
|---|---|---|---|---|---|
| **Recovery rate [n (%)]** | | | | | |
| ITT | 238 (73.5) | 123 (75.9) | 115 (71.0) | 4.9 (-4.70, 14.50) | 0.314 |
| PP | 234 (74.8) | 121 (76.6) | 113 (72.9) | 3.7 (-5.91. 13.31), | 0.454 |
| **Weight gain (g/kg/day ±SD)** | | | | | |
| ITT | 3.14 (2.0) | 3.18 (1.89) | 3.09 (2.05) | 0.10 (-0.33, 0.53) | 0.642 |
| PP | 3.23 (1.92) | 3.25 (1.86) | 3.21 (1.98) | 0.04 (-0.38, 0.47) | 0.838 |
| **MUAC gain (mm/day ±SD)** | | | | | |
| ITT | 0.12 (0.08) | 0.12 (0.07) | 0.11 (0.08) | 0.01 (-0.01, 0.03) | 0.209 |
| PP | 0.12 (0.07) | 0.12 (0.07) | 0.11 (0.07) | 0.01 (-0.01, 0.02) | 0.392 |
| **Height gain (mm/day ±SD)** | | | | | |
| ITT | 0.4 (0.3) | 0.39 (0.26) | 0.36 (0.25) | 0.04 (-0.02, 0.10) | 0.208 |
| PP | 0.38 (0.25) | 0.40 (0.25) | 0.37 (0.25) | 0.03 (-0.02, 0.09) | 0.244 |
| **Recovery time (mean in days ±SD)** | | | | | |
| ITT | 55.6 (26.4) | 54.27 (26.74) | 56.90 (25.99) | -2.64 (-8.40, 3.13) | 0.368 |
| PP | 54.6 (26.3) | 53.52 (26.65) | 55.68 (25.91) | -2.17 (-7.97, 3.64) | 0.466 |

LIBS: local-ingredients-based supplement; CSB+: improved corn-soya blend; g: gram; kg: kilogram; MUAC: mid-upper arm circumference; mm: millimeter; ITT: intention-to-treat; PP: per-protocol.

Data are n (%) when using the chi-square test and mean ± SD and mean difference (95% CI) when using the generalized estimating equation.

## Discussion

This randomized controlled non-inferiority trial compared the efficacy of LIBS and CSB+ in treating MAM. In this study, the ITT, as well as PP analyses, showed that children with MAM

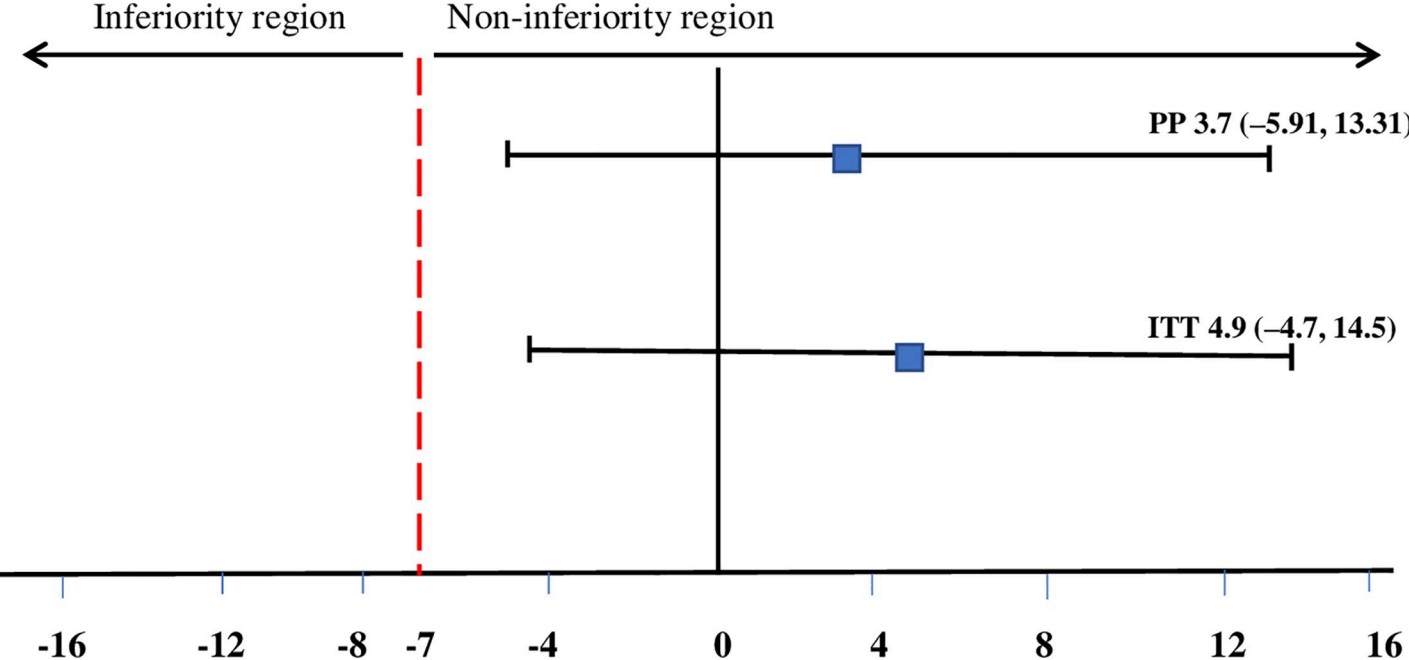

**Fig 2. Difference in recovery rate between LIBS and CSB+ groups.** The recovery rate in the LIBS group was non-inferior compared with the CSB+ group. Numbers are risk differences in recovery rates between groups (%). The dotted line indicates the predefined non-inferiority margin. Abbreviations: LIBS: local-ingredients-based supplement; CSB+: corn-soy blend plus; PP: per-protocol; ITT: intention to treat.

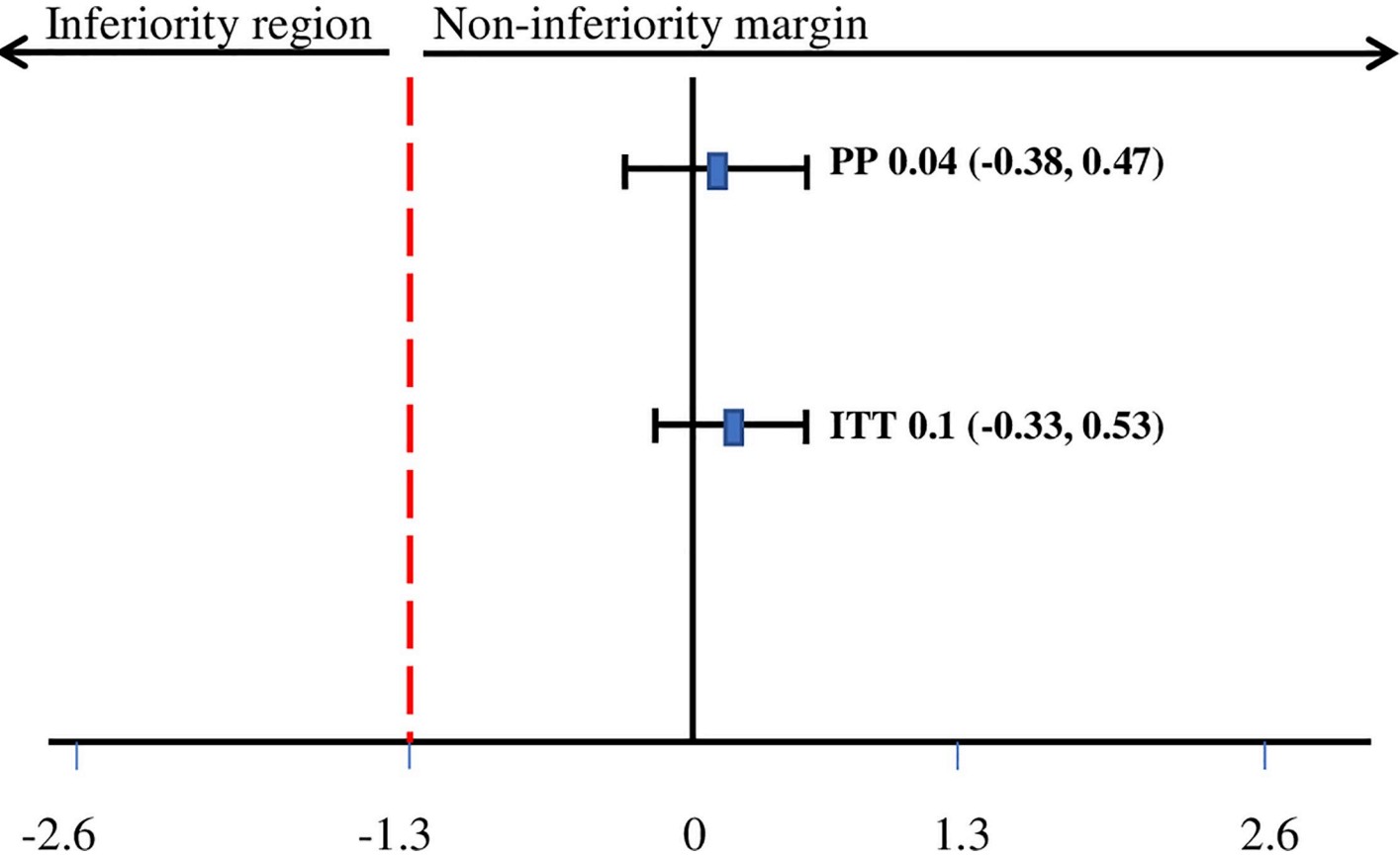

**Fig 3. Difference in average weight gain (g/kg/day) between LIBS and CSB+ groups.** The average daily weight gain in the LIBS group was non-inferior compared with the CSB+ group. Numbers are risk differences in average weight gains between groups (g/kg/day). The dotted line indicates the predefined non-inferiority margin. Abbreviations: LIBS: local-ingredients-based supplement; CSB+: corn-soy blend plus; PP: per-protocol; ITT: intention-to-treat. The generalized estimating equation was used.

who received LIBS were not inferior compared with those who received CSB+ in terms of recovery rate, weight gain recovery time, MUAC gain, and length/height gain. Only the LIBS groups met the Sphere goal of 75% recovery.

The recovery rate of MAM observed in this study was comparable with the recovery rates of MAM with supplementary feeding reported in other studies (67% to 82.3%) [10, 15, 19, 26] and higher than the recovery rates of MAM with child-centered counseling intervention only (57.8%) [26]. The comparability of recovery rate with other studies, the non-inferiority of LIBS compared with CSB+, and the fact that the PP and ITT analyses presented similar results of recovery rate are strong evidence indicating that LIBS is an excellent alternative to CSB+ in treating MAM [27].

Additionally, these results are in line with findings of a study done in Uganda where standard therapeutic food for MAM (CSB+) and a locally produced supplement resulted in a comparable recovery rate among wasted children [15]. By contrast, this finding was inconsistent with a study done in Malawi [20], where the efficacy of CSB++ was compared with that of two RUSFs in treating MAM.

A possible explanation for the comparable recovery rate between LIBS and CSB+ groups could be related to both supplements having a similar energy density, as the energy density found in supplementary food is important for the recovery of MAM children [28]. During

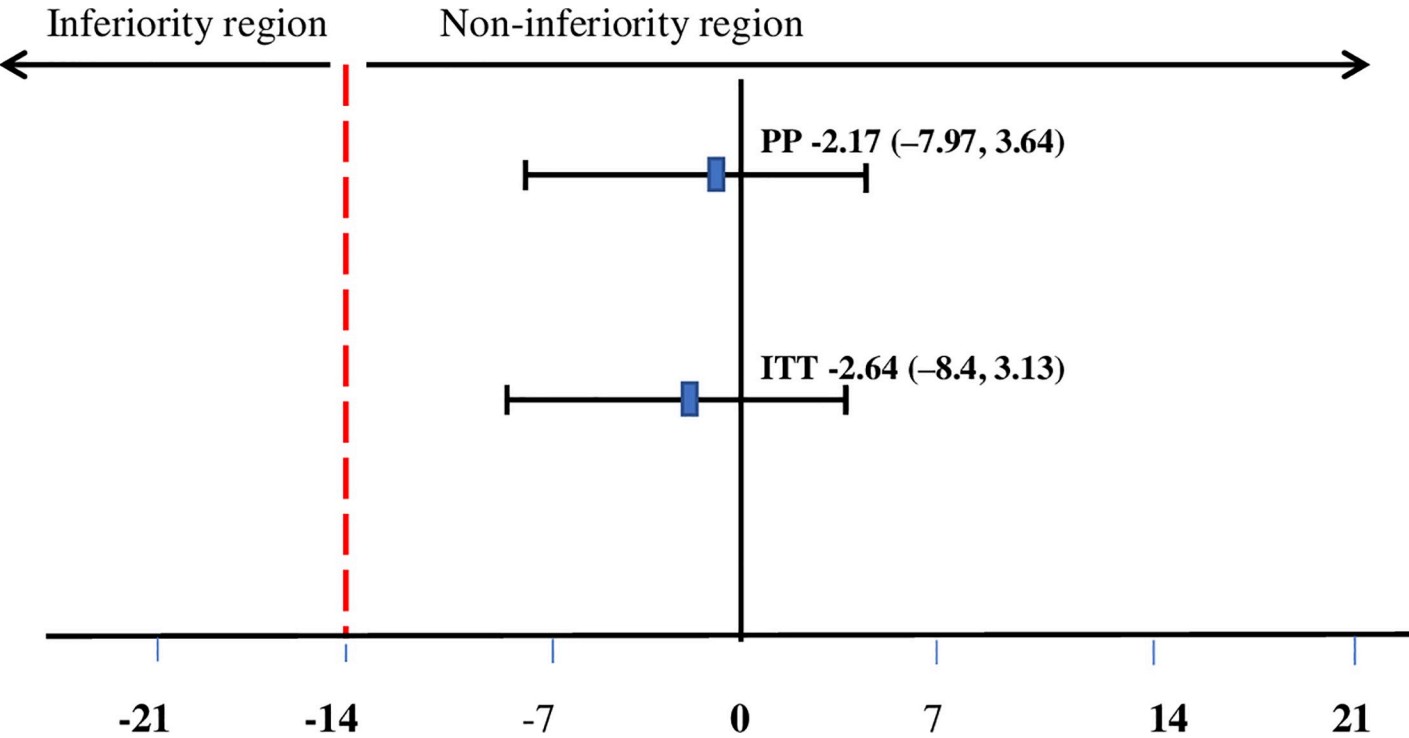

**Fig 4. Difference in recovery time in days between LIBS and CSB+ groups.** The recovery time in the LIBS group was non-inferior compared with the CSB+ group. Numbers are risk differences of recovery time in days between groups. The dotted line indicates the predefined non-inferiority margin. Abbreviations: LIBS: local-ingredients-based supplement; CSB+: corn-soy blend plus; PP: per-protocol; ITT: intention to treat. The generalized estimating equation was used.

malnutrition, energy is needed to enhance catch-up growth and maintain the replacement of the loss of both lean and fat tissue [29].

In this study, the overall rate of loss to follow-up was similar between the LIBS and CSB + groups and met the Sphere target rates for defaulting (≤15%). In the present study, the lost-to-follow-up rate was remarkably low (1.5%), much lower than that of previous studies, which had default rates of 4% to ≥5% [30–32]. This might be due to the weekly follow-up done by the researchers and the daily visits done by food distributors.

CSB+ has been used as a standard therapeutic food for MAM for a long time and is very well accepted, with a renowned feeding protocol in the study area. CSB+ is organoleptically tolerable in various settings [20]. Similarly, LIBS was developed from culturally known and acceptable foods and prepared with a similar appearance to conventional food (CSB+). This could be another explanation for the comparability of the default rate between the two groups.

About 2% of children recruited in this study did not respond to treatment, but continued to lose weight and developed SAM. There was no significant difference in the number of children who developed SAM between the two groups. The reason for the progression to SAM is unknown and might not be related only to household food insecurity, in that it could be explained by the hypothesis that these children had an untreated illness. However, this rate of progression to SAM finding is lower than studies done in Cameroon [10], Malawi [20], and Burkina Faso [26].

This study showed that the average weight gain observed in the LIBS group was not inferior to the CSB+ group. This could be explained by the fact that the total energy provided by the supplements in both groups was comparable. Another explanation might be that both supplements need to be cooked before feeding to the child. The amount of water needed for the

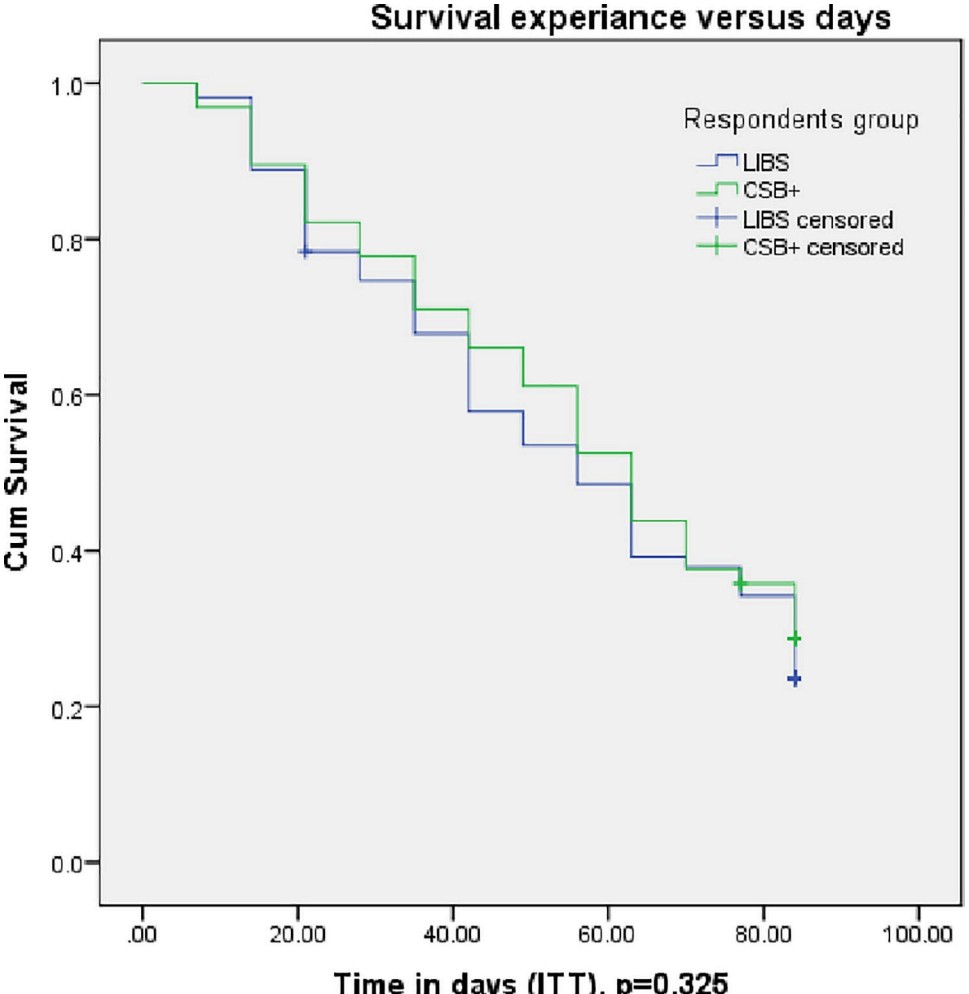

**Fig 5. Recovery of children with moderate acute malnutrition over time in the LIBS and CSB+ arms in ITT analysis.** The recovery of children with MAM over time in the LIBS group was similar to the CSB+ group. Survival analysis, Kaplan-Meier curves and Log-rank tests were used to analyze and describe the data. Abbreviations: MAM: moderate acute malnutrition; LIBS: local-ingredients-based supplement; CSB+: corn-soy blend plus; ITT: intention to treat.

cooking of both supplements was similar in that the mass of supplements fed to children was comparable. The finding also compared favorably with weight gain observed in a study carried out in Malawi [20] and was higher than studies done in Cameroon and Malawi [10, 33].

In this study, the recovery time between the LIBS and CSB+ groups was similar. This finding is supported by other studies, for example, in Cameroon, where the efficacy of CSB+ and RUSF were compared in treating MAM [10], and in Uganda, where the efficacy of sorghum-peanut blend and CSB+ had been compared as therapeutic food for MAM [15]. By contrast, the recovery time was extended compared with a study done in Burkina Faso [26] However, it was within the acceptable Sphere Standards ($\leq$ 90 days) [15].

The length/height gain was comparable between the two groups. This could be explained by the similar amount of protein and zinc found in the LIBS and CSB+. Complementary, supplementary, and therapeutic foods with good quality protein are effective for promoting the growth of children [34]. Similarly, zinc is vital for growth and development and is a growth

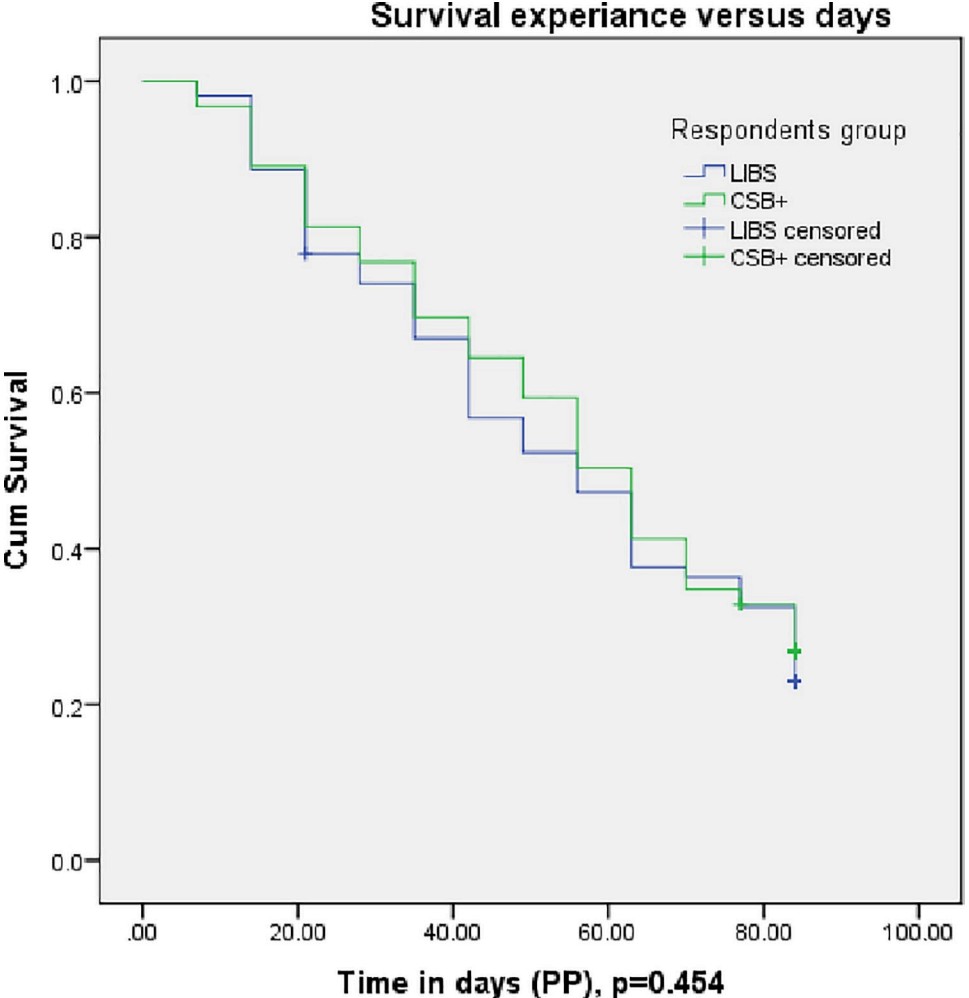

**Fig 6. Recovery of children with moderate acute malnutrition over time in the LIBS and CSB+ arms in PP analysis.** The recovery of children with MAM over time in the LIBS group was similar to the CSB+ group. Survival analysis, Kaplan-Meier curves, and Log-rank tests were used to analyze and describe the data. Abbreviations: MAM: moderate acute malnutrition; LIBS: local-ingredients-based supplement; CSB+: corn-soy blend plus; PP: per-protocol.

type II [29]. It has direct effects on the key hormonal system (IGF-I/GH) that controls growth in the postnatal phase [35]. Our finding was comparable with the study done by Stobaugh [36] and better than studies done by LaGrone et al., Nikièma et al., and Patel et al. [20, 26, 33].

This non-inferiority trial, to the best of our knowledge, was the first study so far using locally developed supplements based on local food ingredients to compare with the standard treatment of MAM (CSB+) in Ethiopia. One of the strengths of this study is that we have used locally available and culturally acceptable food ingredients to formulate the supplement. Another strength is meeting the Sphere standard regarding recovery rate, recovery time, and loss to follow-up rate. The daily home visits by food distributors and weekly home follow-ups by the data collectors allowed us to minimize the lost-to-follow-up rate. However, the study had some limitations. The main limitation of this study was not monitoring the children's overall dietary intake during the treatment and not having a non-supplemented control group. Thus, we could not measure the contribution of the home diet. Another limitation of this study was that we did not test the shelf life of the newly developed LIBS product. The shelf life

of the product is critical if this product is to be used in the future as therapeutic food for treating children with MAM.

## Conclusion

The present study showed that LIBS was non-inferior to conventional food (CSB+) in treating MAM, and both supplements were relatively successful for treating MAM in children aged 6 to 59 months. The finding from this study provides an experimental indication that LIBS can be used as an alternative to CSB+ in the management of MAM in Ethiopia. Thus, the potential for scaling up the use of LIBS should be promoted. Further studies should also examine concerns such as the product's shelf life, and its sustainability, and cost-effectiveness. Community perceptions of the LIBS as a treatment for MAM should also be tested. The evidence drawn from this study will be shared with the public and with policymakers as LIBS has the potential to increase recovery from MAM and decrease the burden of malnutrition.

## Supporting information

**S1 File. CONSORT checklist.**
(DOC)

**S2 File. Fully anonymized database in SPSS format.**
(SAV)

## Acknowledgments

We thank all the participating children and their mothers or caregivers who consented to offer their time and contribute throughout the study time. We are grateful to data collectors, supervisors, food distributors, healthcare providers, and all others who facilitated our work.

## Author Contributions

**Conceptualization:** Debritu Nane, Anne Hatløy, Bernt Lindtjørn.

**Data curation:** Debritu Nane, Anne Hatløy.

**Formal analysis:** Debritu Nane.

**Funding acquisition:** Debritu Nane, Bernt Lindtjørn.

**Investigation:** Debritu Nane, Anne Hatløy, Bernt Lindtjørn.

**Methodology:** Debritu Nane, Anne Hatløy.

**Project administration:** Debritu Nane, Bernt Lindtjørn.

**Resources:** Debritu Nane, Bernt Lindtjørn.

**Software:** Debritu Nane.

**Supervision:** Debritu Nane, Anne Hatløy, Bernt Lindtjørn.

**Validation:** Debritu Nane, Anne Hatløy, Bernt Lindtjørn.

**Visualization:** Debritu Nane, Anne Hatløy.

**Writing – original draft:** Debritu Nane.

**Writing – review & editing:** Debritu Nane.

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
