## [Decision Letter · Decision Letter 0]

20 Jan 2021

PONE-D-20-29119

A local-ingredients-based supplement is an alternative to corn-soy blends plus for treating moderate acute malnutrition among children aged 6 to 59 months: A randomized controlled non-inferiority trial in Wolaita, Southern Ethiopia

PLOS ONE

Dear Dr. Nane,

Thank you for submitting your manuscript to PLOS ONE. After careful consideration, we feel that it has merit but does not fully meet PLOS ONE’s publication criteria as it currently stands. Therefore, we invite you to submit a revised version of the manuscript that addresses the points raised during the review process.

We look forward to receiving your revised manuscript.

Kind regards,

Seth Adu-Afarwuah

Academic Editor

PLOS ONE

Journal Requirements:

2. Thank you for submitting your clinical trial to PLOS ONE and for providing the name of the registry and the registration number. The information in the registry entry suggests that your trial was registered after patient recruitment began. PLOS ONE strongly encourages authors to register all trials before recruiting the first participant in a study.

1) your reasons for your delay in registering this study (after enrolment of participants started);

2) confirmation that all related trials are registered by stating: “The authors confirm that all ongoing and related trials for this drug/intervention are registered.

Reviewers' comments:

Reviewer's Responses to Questions

**Comments to the Author**

1. Is the manuscript technically sound, and do the data support the conclusions?

Reviewer #1: Partly

Reviewer #2: Yes

Reviewer #3: Yes

Reviewer #4: Yes

2. Has the statistical analysis been performed appropriately and rigorously? 

Reviewer #1: No

Reviewer #2: Yes

Reviewer #3: Yes

Reviewer #4: Yes

3. Have the authors made all data underlying the findings in their manuscript fully available?

Reviewer #1: Yes

Reviewer #2: Yes

Reviewer #3: Yes

Reviewer #4: Yes

4. Is the manuscript presented in an intelligible fashion and written in standard English?

Reviewer #1: Yes

Reviewer #2: Yes

Reviewer #3: Yes

Reviewer #4: Yes

5. Review Comments to the Author

Reviewer #1: The manuscript entitled ‘’A local-ingredients-based supplement is an alternative to corn-soy blends plus for treating moderate acute malnutrition among children aged 6 to 59 months: A randomized controlled non-inferiority trial in Wolaita, Southern Ethiopia” with the aim to compare a local-ingredients-based supplement (LIBS) with the standard corn-soy blend plus (CSB+) in treating MAM among children aged 6 to 59 months to test the hypothesis that the recovery rate achieved with LIBS will not be more than 7% worse than that achieved with CSB+.

The manuscript requires further improvement based on the following comments.

Page 8 Line 160, the sentence ‘’The two research teams included two supervisors, 12 data collectors and 18 food distributers’ to be revised and clearly the numbers for each team.

Sample size calculation

Page 9, 1 or 2-tailed test to be stated.

Randomization

Page 10 Allocation concealment to be stated.

Page 10 Line 202-206 the steps/process not clear and requires revision.

Statistical analysis

Page 14, Line 279, the version and publisher name for SPSS and STATA to be stated.

Page 14 Line 283, the word student’s to be omitted from Student’s chi-square test

Page 14 Line 281-286, for statistical tests, 1 or 2-tailed test to be stated.

The registration of the clinical trial to be stated.

Page 15 Line 303, ‘’No serious adverse reactions were detected’’ to be placed in results section.

The statistical tests which were mentioned in the statistical analysis section to be clearly indicated/denoted in the table(s)/ figure(s)/text results

Results

Based on CONSORT guidelines, all statistical analyses at baseline for group comparison (Table 2) to be avoided.

Page 14 Line 283, the sentence ‘’medians with the use of the Mantel-Haenszel test’ not clear and requires revision.

Page 17 Table 2, decimal points to be standardized. Likewise in other sections where applicable.

Page 18 Line 343-353, more than 1 decimal point for p value to be provided. Decimal points for 95%CI to be standardized.

Page 19 Table 3, baseline data/findings to be presented apart from results in Week 12 and followed by difference. Both ITT and PP results to be presented.

Page 20 Line 383-387, separate diagram to be presented to show both ITT and PP results.

Page 20 Line 383, the sentence ‘’The difference in recovery time was not statistically significant (p=0.8)’not clear and requires revision.

Figure 1, the background dark colour to be removed.

Figure 1 and 4 The ITT and PP analysis to be provided. Figure 4, p value for group comparison to be stated.

For the Figure 2, 3, 4 title, description of character A, B, C and D to be clearly denoted in the figures. Otherwise the labelling A, B, C, D to be omitted.

Reviewer #2: This is an important study and I appreciate the opportunity to have reviewed it. The o is excellent.

Please refer to the attached marked up document for specifics on revisions.

Major recommendations:

1. do a thorough job of defining MAM - it is incomplete

2. do a thorough job of specifying what is meant by sugar, oil, vegetable oil - these are general terms

3. focus the Introduction - some of the info here is better combined and condensed into the Discussion

Reviewer #3: Overall comment

This is a well-designed study and makes a strong case for a potential use of a supplementary food product from local ingredients for management of moderate acute malnutrition in Ethiopian context.

Find some specific comments below:

Abstract

- Prevalence of wasting (12.5%)—does it include SAM and MAM?

- The 7% recovery rate with LIBS—how was this determined, random or evidence based? Why not 5% or 10% - please provide some description if not already provided in the main text

- Treatment time length was not specified in the abstract (how long did the treatment last?)

- Conclusion sounds a bit too strong just for a first of such a study and in a just a small locality…

Introduction

- Line 111-112: Non inferiority trial of local product at least as effective as the standard product (CSB+) (does this align with the hypothesis that local product efficacy no less 7% recovery rate?)

- Last paragraph (line 114-117): “Besides this, adequate research has not been conducted in Ethiopia on LIBS, even though it has the potential to contribute to treating MAM. The evidence drawn from this study will be shared with the public, policymakers, and scholars, as LIBS has the potential to increase recovery from MAM and decrease the burden of malnutrition.” – it may be appropriate to briefly mention local research on supplementary foods using local ingredient for treatment of acute malnutrition.

- Also the discussion re the benefits of LIBS and its promotion to decision makers should come in the discussion/conclusion section.

Material and methods

- Describe the specific kebele’s the study subjects were recruited from

- Exclusion criteria – how did you apply the criteria (who determined if the child was ill/had medical complication, or was SAM case, etc.) please describe.

- Why is the objective of the study in materials & methods section (line 147-151) – please move to the end of introduction section.

- Line 151: how is the 7% margin determined? What would be acceptable margin, 10%, 5% or 3%, etc.?

- Line 166 – 168: What were the supplementary food distribution techniques, cooking process and feeding procedures? Who are these people?

- Line 188: “skilled data collectors” – skilled in what? Either describe that or better to say “trained data collector”

- Line 189: “randomly selected kebeles”- why randomly select kebeles? Does it have any relevance for your objective and type of study?

- Line 247-249: The composition of the LIBS product (expressed per 100 g) and the ingredient amounts do not match up (“30 g of pumpkin seed, 25 g of peanut grain,

- 249 20 g of amaranth grain, 15 g of flaxseed, 10 g of emmer wheat, 25.2 g of sugar and 8 ml of oil.” Is more than the 100 g of LIBS)…

Results

- Lines 301-302 & 334: if literacy rate was low in the area, how did you manage to obtain “written informed consent”?

Discussions

- Line 447: “The length/height gain was not comparable between the two groups.” ….was or was not comparable? I think this might be a typo error.

Conclusion

- Your conclusion here and in the abstract seem a bit different (in the abstract, you make a stronger claim of LIBS being an “effective alternative” to CSB+). Since your study has not shown feasibility of cost, product shelf life, reliability of local ingredient supplies, etc. you might want to tone down your conclusions.

Reviewer #4: This was a very interesting, well-written paper describing the possible use of locally produced food supplements in Ethiopia. It is clear that the CONSORT statements were used to execute the study and write the paper, even if it is not officially declared. Data analyses included per-protocol and intention-to-treat results.

I only have a few comments:

1. Line 27: it is not clear why 7% recovery rate was used as a cut-off value to make conclusions regarding non-inferiority. Please elaborate on this. In the abstract as well as the text.

2. Line 99: Did the LIBS contain vitamin and mineral supplements similar to the CBS+?

3. Line 111: replace demonstrate with determine.

4. Line 250: the text read: this supplement..... does that mean one serving contained the mentioned supplements or was this daily contribution to total consumption per day? Please elaborate on this.

Line 402: Provide the recovery rates of "MAM with child-centred counselling interventions only" to give a clear picture of the impact of the supplementary feeding.

General comments:

1. Was there any part of the study that focussed on relapse?

2. Did the fieldworkers actually observe the consumption of the supplements at all times and of all meals? What was the possibility of sharing amongst other family members? I understand that provision was made for twins, but what about other family members?

3. Was the actual amount consumed measured or only the amount provided to the child? In other words, did the fieldworkers weigh food before and after consumption? The lack of additional food consumption data was a serious limitation so it is not clear if the children consumed only the supplement or additional family food.

4. The randomisation of the included children was described in detail, but it is unclear how the Kebeles were selected. The paper (line 134) only states “selected kebeles” and line 189: randomly selected kebeles. Please elaborate on this as it might influence the representativeness of the study participants.

5. It is unclear who knew the allocation of the two groups since the food distributors were also blinded. If the producers knew the difference between the two, when was allocation revealed? It is also not clear how many portion participants consumed per day (all at one meal, or spread throughout the day?)

6. PLOS authors have the option to publish the peer review history of their article (what does this mean?). If published, this will include your full peer review and any attached files.

Reviewer #1: No

Reviewer #2: **Yes: **Michael I Lindinger

Reviewer #3: **Yes: **Getahun Ersino (also Getahun Lombamo)

Reviewer #4: **Yes: **MJ Lombard

---

## [Author Response · Author response to Decision Letter 0]

8 Mar 2021

PLOS ONE

07.03.21

Dear Editor 

Thank you for allowing us to submit a revised draft of our manuscript titled ‘A local-ingredients-based supplement is an alternative to corn-soy blends plus for treating moderate acute malnutrition among children aged 6 to 59 months: A randomized controlled non-inferiority trial in Wolaita, Southern Ethiopia’ to PLOS ONE journal. 

We are thankful to the editor and reviewers for their constructive comments which enabled us to revise the manuscript. We have tried to improve the manuscript according to the recommendations raised by the Editor and reviewers. 

We have highlighted the changes within the manuscript. For the reference of pages and line numbers, we used a clean version of the manuscript. In this letter, the reviewers’ and Editor’s comments are written in bold format and our point-by-point responses are written in normal font. 

Yours sincerely,

Debritu Nane

Corresponding Author

 

Comments and questions from the Academic editor

Thank you for submitting your clinical trial to PLOS ONE and for providing the name of the registry and the registration number. The information in the registry entry suggests that your trial was registered after patient recruitment began. PLOS ONE strongly encourages authors to register all trials before recruiting the first participant in a study.

1) your reasons for your delay in registering this study (after enrolment of participants started)

Regarding the reason for retrospective trial registration, we applied for the registration of clinical trial to Pan-African Clinical Trial Registration on 11/9/2018. While applying for the registration, we wrote that 27/7/2018 as the anticipated date of participant recruitment. The mistake was made while filling out the registration form. However, our effort to correct the date after the submission was not accepted by the contact person. We started the house-to-house screening (baseline screening) of children for MAM using only MUAC on August 21, 2018. The actual screening based on the eligibility criteria and using both MUAC and WHZ started on August 27, 2018 and continued until September 20, 2018. We didn’t include this information in the revised manuscript since it looks a bit detailed. 

2) Confirmation that all related trials are registered by stating: “The authors confirm that all ongoing and related trials for this drug/intervention are registered.

Response: We accepted the comment and we included the statement in the revised manuscript. It now reads; 

“The authors confirmed that all ongoing and related trials for these food supplements were registered.” The revision is found in the revised manuscript on page 7, line 127 as all ongoing and related trials for this intervention were registered.

3) We note that you have indicated that data from this study are available upon request. PLOS only allows data to be available upon request if there are legal or ethical restrictions on sharing data publicly. For information on unacceptable data access restrictions, please see http://journals.plos.org/plosone/s/data-availability#loc-unacceptable-data-access-restrictions.

Response: The anonymized data has now been uploaded as a supplementary file and addressed in the revised cover letter.

Reviewers' comments and questions: Reviewer #1

Thank you very much for your helpful comments and questions. According to the comments and questions, we included the information that describes the numbers of research teams, allocation concealment, version and publisher name for softwares, statistical tests, trial registration, both ITT and PP results, tables, and figures that showed both ITT and PP results and p values for group comparison of recovery time. We omitted statistical analysis used for baseline comparison of groups, labeling used to describe figures (A, B, C, D), and words that introduce confusion. 

The manuscript requires further improvement based on the following comments.

Page 8 Line 160, the sentence ‘’The two research teams included two supervisors, 12 data collectors and 18 food distributers’ to be revised and clearly the numbers for each team.

Response: We accepted the comment and put the numbers for each team. The revision is found on page 9, lines 170 to171. It now reads as ; “There were two research teams, each team consisted of one supervisor, six data collectors and eighteen food distributors (selected females living in the study area)”.

Sample size calculation

Page 9, 1 or 2-tailed test to be stated.

Response: We agree. We included the required information as follows. The sample size was calculated based on 80% power of the test, 7% margin of non-inferiority, and assuming a recovery rate with CSB+ of 67%. Considering the above assumptions, a total of 324 children (162 subjects per study group), allowing for a 10% withdrawal rate were required to be sure that the lower limit of a two-sided 95% confidence interval (CI) was above the -7% margin of non-inferiority (pages 10, lines 190-195).

Randomization

Page 10 Allocation concealment to be stated.

Response: We accepted the comment and revised it as follows. A computer-generated, randomization list that contained codes was prepared using random allocation software. The allocation ratio was 1:1. Mothers selected an envelope containing coded numbers that matched with one of the two supplementary foods. This revision is found on page 11, lines 220 to 223.

Page 10 Line 202-206 the steps/process not clear and requires revision.

Response: We accepted the comment and revised the steps by omitting the words that introduce confusion (page 11, lines 220 to 223 and 227 to 229). 

Statistical analysis

Page 14, Line 279, the version and publisher name for SPSS and STATA to be stated.

Response: We added the publisher name for SPSS as 20 (IBM, Chicago, IL, U.S.A.) and for STATA as 15 (StataCorp LLC). This revision is found on page 15, line 299.

Page 14 Line 283, the word student’s to be omitted from Student’s chi-square test

Response: We agree. All statistical analyses at baseline for group comparison are now removed. The revision is found on page 15, lines 301 to 302. 

Page 14 Line 281-286, for statistical tests, 1 or 2-tailed test to be stated.

Response: We accepted and included the comment in the revised manuscript as “A difference in the estimated proportion between the groups along with 95 % CI, a two-tailed test was estimated to infer non-inferiority”. Our trial is actually tests against a one-sided hypothesis but the decision of non-inferiority is based on a two sided 95% confidence interval even if we deal with the lower limit of the interval. The revision is found on page 15, lines 304 to 307.

The registration of the clinical trial to be stated.

Response: We have now indicated this and included it in the method section as: This trial was registered at Pan-African Clinical Trial Registration as PACTR201809662822990. The authors confirmed that all ongoing and related trials for these food supplements were registered. The amendment is found on page 7, lines 127 to 129.

Page 15 Line 303, ‘’No serious adverse reactions were detected’’ to be placed in results section.

Response: Agree. The sentence, “No serious adverse reactions were detected’’ was deleted from the method section and moved to result section. This edition is found on page 16, lines 324 to 325.

The statistical tests which were mentioned in the statistical analysis section to be clearly indicated/denoted in the table(s)/ figure(s)/text results

Response: We indicated the statistical tests in the description parts of tables, figures, and text results. 

The amendments are as follows. Fisher’s exact test and chi-square test for dichotomous variables and generalized estimating equation for continuous variables were used to analyze baseline differences between groups. This is represented under table 2 page 18, lines 359 to 360. We also added the statistical tests that used to compare the outcome variables between groups. The revisions are found in the manuscripts on page 19, line 362, under the description of Fig.3 page 20, line 400, under the description of Fig.4 page 21, lines 412 to 413 and under table 3, page 21 lines 417 to to 418. Survival analysis, Kaplan-Meier curves, Log-rank tests were used to analyze the recovery of children with moderate acute malnutrition over time between groups in ITT as well as PP analysis. This is denoted under the description of Fig 5 page 22, lines 424 to 425 and under the description of Fig 6 page 22, lines 431 to 432.

Results

Based on CONSORT guidelines, all statistical analyses at baseline for group comparison (Table 2) to be avoided.

Response: All stated statistical analyses about baseline group comparison are now deleted from the text. This edition is found on page 15, lines 301 to 302.

Page 14 Line 283, the sentence ‘’medians with the use of the Mantel-Haenszel test’ not clear and requires revision.

Response: The sentence has now been omitted from the text. This is found on page 15, line 301 to 302.

Page 17 Table 2, decimal points to be standardized. Likewise in other sections where applicable.

Response: We accepted the comment and revised/standardized the decimal points of table 2. We used three decimal digits to report p values. The table is found in the revised manuscript on page 18, line 357.

Page 18 Line 343-353, more than 1 decimal point for the p-value to be provided. Decimal points for 95%CI to be standardized. 

Response: We revised the paragraph based on the comment. we have now provided three decimal points to P values. and two decimal points to 95 % CIs and mean differences. 

The revised paragraph is found in the revised manuscript on page 19, lines 366 to 370.

Page 19 Table 3, baseline data/findings to be presented apart from results in Week 12 and followed by difference. Both ITT and PP results to be presented.

Response: We accepted the comment and included both ITT and PP results, proportion/mean differences with their 95% CI, and p values in the table. The table is found in the revised text, page number 21, line 414. Here is the edited table. 

The table is found in the revised text, page number 21, line 414. The edited table is included in the currently uploaded response to reviewer letter. 

Page 20 Line 383-387, the separate diagram to be presented to show both ITT and PP results.

Response: We have now added another figure (Fig 4) that compares the recovery time between LIBS and CSB+. The non-inferiority of the LIBS group compared with CSB+ is indicated in the figure. This figure (Fig 4) has been uploaded as a figure file. The description of this figure is found on page 21, lines 408 to 413. We have uploded the figure as a figure file.

Page 20 Line 383, the sentence ‘’The difference in recovery time was not statistically significant (p=0.8)’not clear and requires revision.

Response: We accepted the comment and revised it as: There is no difference in recovery time between the LIBS and CSB+ groups in ITT analysis p=0.368 and in PP analysis p= 0.466. This revision is found on page 21 lines 420 to 421. 

Figure 1, the background dark colour to be removed.

Response: We agree. The background dark color has been removed from the figure.

Figure 1 and 4 The ITT and PP analysis to be provided. Figure 4, p value for group comparison to be stated.

Response: We included all recruited children in ITT analysis whereas, children who were lost to follow-up or who were referred to the severe acute malnutrition clinic excluded in PP analysis. We included the above information into the figure1 (trial profile). We have also added this in the description of figure 1 (Pages 16 and 17, lines 335 to 338). The figure (Fig 1) has been uploaded as a figure file.

Regarding figure 4 (the mean recovery time), another figure was added that describes the difference in recovery time in PP analysis. Fig 4 changed to Fig 5 and describes the mean recovery time between groups in ITT analysis whereas Fig 6 describes the mean recovery time between groups in PP analysis with their respective p-values. The description of Fig 5 is found in the revised manuscript on page 22, lines 422 to 427. The description of Fig 6 is found in the revised manuscript on page 22, lines 428 to 433. Both figures have been uploaded as a figure files.

For the Figure 2, 3, 4 titles, description of character A, B, C and D to be clearly denoted in the figures. Otherwise the labelling A, B, C, D to be omitted.

Response: The labellings (A, B, C, D) are now omitted from figures descriptions. The revision of figure 1 description is found on pages 16 to 17, lines 326 to 338; page 19, lines 376 to 380 (descriptions of Fig 2); page 21, lines 408 to 413 (descriptions of Fig 4) and page 22, lines 422 to 427 (descriptions of Fig 5).

Reviewer #2 

This is an important study and I appreciate the opportunity to have reviewed it. The o is excellent. Please refer to the attached marked-up document for specifics on revisions.

Based on the comments and questions, we described the definition of terms (wasted, cereal, and MAM), the specification of sugar and oil, and the amount of carbs and ash. We have added one paragraph that explains the aspect of how impaired growth and development is considered as acute condition. We also revised the introduction part. We reduced the length and omitted some statements from the introduction part. The ethical approval part is put at the beginning of the method section.

Responses for the comments specified/tracked in the manuscript

- Regarding the comments to define the term “wasted”, it was defined as acutely malnourished throughout the document. 

- For the comment you raised in the method part of the abstract to specify sugar and oil, we included the the revised sentence in the manuscript. It now reads; 125g of LIBS with 8 ml of sunflower oil. (page 2, lines 32 to 33).

- To respond to the comment indicated on page 4, lines 55-56 (explain what aspect of impaired growth and development is acute), the information is added to the revised text on page 4, lines 53 to 57. The references we used to write the added paragraph are found in the last page of this letter (1, 2).

- Regarding the question mentioned in the manuscript on page 4, line 63 (what acute means in terms of nutrient delivery), the newly added information describes it and is found in the revised manuscript on page 4, lines 53 to 57.

- The specification of vegetable oil as sunflower oil and sugar as cane sugar was made throughout the document. These are found on page 6, line 101; page 8 and 9, lines 159 to 160; page 13, lines 266-, 269, 271 and page 14, line 282.

- To respond to the question raised on page 6 line 91as: What cereals and grains are used in the formulation of CSB++, we described the cereals like corn and soybean. This revision is found in the revised manuscript on page 5, lines 80 to 81.

- The last comment of the introduction part was “introduction needs better focus; reduce length at least 30%; restrict to the introduction, not discussion.” Based on the comment, we restricted the introduction by removing unnecessary statements. We now removed some sentences from the introduction part and moved to the discussion part. These revisions are found in the revised manuscript on pages 4 to 6, lines 50 to 103 and page 26, lines 515 to 517.

- The ethical statement was added at the beginning of the materials and methods part. (Page 7, lines 117 to 126)

- The specification of vegetable oil and sugar was made in the revised manuscript on page page 6, line 101; page 8 and 9, lines 159 to 160; page 13, lines 267, 271 and page 14, line 282.

- The amount of carbohydrate and ash found in both supplements was added to the revised manuscript. These revisions are found on page 13, lines 270 and 271; 14 line282 in table 1.

- Regarding the comment indicated on page 12, line 251 to specify the term flour, we now deleted the unnecessary word (flour) from the sentence. The amendment is found in the revised text on page 13, line 270.

Major recommendations:

1. do a thorough job of defining MAM - it is incomplete

Response: We defined moderate acute malnutrition (MAM) as weight-for-height z-score (WHZ) between -3 and -2 standard deviations and/or mid-upper arm circumference (MUAC) of between ≥11.5 cm and <12.5 cm, without bipedal edema (Page 4, lines 57 to 60).The reference is included in the manuscript. 

2. do a thorough job of specifying what is meant by sugar, oil, vegetable oil - these are general terms. 

Response: We accepted and specified the vegetable oil as sunflower oil and sugar as cane sugar. The revisions are found in the manuscript on page page 6, line 101; page 8 and 9, lines 159 to 160; page 13, lines 266-269, 271 and page 14, line 282.

3. focus the Introduction - some of the info here is better combined and condensed into the Discussion

Response: Based on the comment, the revision was made to the introduction. This was indicated in the above response. The revisions are found in the revised manuscript on page on pages 4 to 6, lines 50 to 103 and page 26, lines 515 to 517.Reviewer #3: Overall comment

This is a well-designed study and makes a strong case for a potential use of a supplementary food product from local ingredients for management of moderate acute malnutrition in Ethiopian context.

Find some specific comments below:

According to the comments and questions, we described how we select the 7% non-inferiority margin, the duration of treatment, the names of kebeles, how to select kebeles, exclusion criteria, supplementary food distribution techniques, cooking process, and feeding procedures. We have also revised the conclusion and the nutrients description of LIBS. Some unnecessary sentences were removed from the text. 

Abstract

Prevalence of wasting (12.5%)—does it include SAM and MAM?

Response: It was about acute malnutrition (SAM and MAM) but we do not want to keep the sentence in the text and now removed it from the text since it introduces confusion. This was indicated on page 2, lines 21 to 22.

The 7% recovery rate with LIBS—how was this determined, random or evidence based? Why not 5% or 10% - please provide some description if not already provided in the main text

Response: The pre-specified non-inferiority margin (7%) was based on evidence. We added the information into the text on how we determined the 7% non-inferiority margin. Here is the explanation.

The previous studies show the children who received CSB had a recovery rate from MAM of 67% and the recovery rate from MAM without any treatment was 54%, hence the difference is 13%. We divided 13% by two and got the number approaching 7%. Thus the non-inferiority margin was specified considering that the LIBS group could have at least a 7% higher recovery rate than in a group with no supplementary food (placebo). This was based on the recommendations of selecting a non-inferiority margin. This text was added to the revised manuscript. The revision is found in the manuscript on page 10, lines 196 to 202. The references we used are listed at the end of this paper and added in the revised manuscript (3,4,5).

Treatment time length was not specified in the abstract (how long did the treatment last?)

Response: The duration of treatment was 12 weeks. This information was included in the abstract on page 2, lines 33 to 34.

Conclusion sounds a bit too strong just for a first of such a study and in a just a small locality

Response: We tried to answer the research question by indicating the similar effect of LIBS and CSB+ in treating MAM. The term “as effective as” looks like a strong phrase thus; we changed it to “can be an alternative.” The edition is found on page 3, line 44.

Introduction

Line 111-112: Non inferiority trial of local product at least as effective as the standard product (CSB+) (does this align with the hypothesis that local product efficacy no less 7% recovery rate?)

Response: We accepted the comment and revised it as this study aims to evaluate if children aged 6 to 59 months suffering from MAM, treated with 125.2 g of LIBS and 8 ml of sunflower oil/day, for 12 weeks, would not have an inferior recovery rate compared to similar children treated with 150 g CSB+ flour and 16 ml of oil/day (with a 7% margin of non-inferiority). This sentence is found in the manuscript on page 6, lines 100 to 103.

Last paragraph (line 114-117): “Besides this, adequate research has not been conducted in Ethiopia on LIBS, even though it has the potential to contribute to treating MAM. The evidence drawn from this study will be shared with the public, policymakers, and scholars, as LIBS has the potential to increase recovery from MAM and decrease the burden of malnutrition.” 

it may be appropriate to briefly mention local research on supplementary foods using local ingredient for treatment of acute malnutrition.

Response: We accepted the comment and revised the sentence. See the revised manuscript on page 6, line 99.

Also the discussion re the benefits of LIBS and its promotion to decision makers should come in the discussion/conclusion section.

Response: We agree. The sentence (The evidence drawn from this study will be shared with the public, policymakers, and scholars, as LIBS has the potential to increase recovery from MAM and decrease the burden of malnutrition) was moved to the conclusion part and found in the revised manuscript on page 26, lines 515 to 517.

Material and methods

Describe the specific kebele’s the study subjects were recruited from

Response: The names of randomly selected kebeles found in Damot Pulassa district were included in the revised manuscript. (Page 11, lines 206 to 207).

Exclusion criteria – how did you apply the criteria (who determined if the child was ill/had medical complication, or was SAM case, etc.) please describe.

Response: The children were assessed for SAM by trained research teams and health extention workers. Medical complications of children were assessed by the senior pediatric nurse working in a “below five years clinic”. This description is found on page 8, lines 151 to 153.

Why is the objective of the study in materials & methods section (line 147-151) – please move to the end of introduction section.

Response: This was written based on the CONSORT Statement 2006 - Checklist for Non-inferiority and Equivalence Trials that recommends writing the objectives and hypothesis under the method section but we accepted the comment and moved it to the end of introduction part. The revision is found in the revised manuscript on page 6, lines 100 to 103

Line 151: how is the 7% margin determined? 

Response: We described how to select and why we select the 7% non-inferiority margin in the above comment. The explanations regarding the determination of non-inferiority margin are found in the manuscript on page 12, lines 223 to 228. 

What would be acceptable margin, 10%, 5% or 3%, etc.?

Response: The noninferiority margin has to be determined based on the evidence available on the efficacy of the standard treatment compared with placebo (5,6). We based on the recovery rate from MAM with CSB+ and with no supplementary food (placebo). The way how we selected the 7% non-inferiority margin is explained above. 

Line 166 – 168: What were the supplementary food distribution techniques, cooking process and feeding procedures? Who are these people?

Response:The food distributors are selected females living in the study area. They were trained for five days on daily supplement distribution, cooking of the supplementary foods as a porridge, and feeding of a child. The porridge was made by diluting 150g of CSB+ with 600 ml water and cook for 10 to 15 minutes whereas the 125.2g of LIBS with 400 ml water and cook for 10 to 15 minutes. The food distributors checked for and measured the amount of supplements consumed by the children. In case of not consuming all the prepared porridge, re-feeding was done. The descriptions are found on page 9 line 171, 177 to 179 and page 14 lines 277 to 281. 

Line 188: “skilled data collectors” – skilled in what? Either describe that or better to say “trained data collector”

Response: We accepted the comment and now rephrased it as the trained data collectors. The revised sentence is found on page 11, line 205.

Line 189: “randomly selected kebeles”- why randomly select kebeles? Does it have any relevance for your objective and type of study?

Response: Damot Pulassa district (woreda) was purposely selected based on a consideration of the high level of food insecurity, high level of child malnutrition, and access to transportation. All the kebeles in the district had a similar admission loads to the management program of MAM, for practical reasons, six out of the 23 kebeles were randomly selected for the study. Children aged 6 to 59 months with MAM, living in selected kebeles of Damot Pulassa district were included in the study. The revision is found on page 8, lines 143 to 148.

Line 247-249: The composition of the LIBS product (expressed per 100 g) and the ingredient amounts do not match up (“30 g of pumpkin seed, 25 g of peanut grain,

- 249 20 g of amaranth grain, 15 g of flaxseed, 10 g of emmer wheat, 25.2 g of sugar and 8 ml of oil.” Is more than the 100 g of LIBS)…

Response: We revised the description of nutrients. The amount of sugar was now added to the total amount of LIBS and written as 125.2 g of LIBS with 8 ml of sunflower oil. The composition of LIBS was: 30 g of pumpkin seed, 25 g of peanut grain, 20 g of amaranth grain, 15 g of flaxseed, 10 g of emmer wheat and, 25.2 g of cane sugar and with 8 ml of sunflower oil. The description is found in the manuscript on page 13, lines 266 to 269.

Results

Lines 301-302 & 334: if literacy rate was low in the area, how did you manage to obtain “written informed consent”?

Response: The literacy rate in the area was low. To manage this issue, we used both verbal (getting thumb mark after reading the information) and written consent. This explanation is found on page 7, line 123.

Discussions

Line 447: “The length/height gain was not comparable between the two groups.” ….was or was not comparable? I think this might be a typo error.

Response: It is a typing error and edited. The revision is found on page 24, line 487.

Conclusion

Your conclusion here and in the abstract seem a bit different (in the abstract, you make a stronger claim of LIBS being an “effective alternative” to CSB+). Since your study has not shown feasibility of cost, product shelf life, reliability of local ingredient supplies, etc. you might want to tone down your conclusions.

Response: In the conclusion part of the abstract, the term “effective alternative” changed to “can be used as an alternative”. This change might down the tone. The edition is found on page 3, line 44.

Reviewer #4:

This was a very interesting, well-written paper describing the possible use of locally produced food supplements in Ethiopia. It is clear that the CONSORT statements were used to execute the study and write the paper, even if it is not officially declared. Data analyses included per-protocol and intention-to-treat results.I only have a few comments:

Based on the comments and questions, we clarified that how we select the 7% margin of error, how we serve the children with the supplement, the responsibility of food distributors related to the supplement consumption, how family sharing was managed, selection of kebeles, and how we allocate the children into groups. 

1. Line 27: it is not clear why 7% recovery rate was used as a cut-off value to make conclusions regarding non-inferiority. Please elaborate on this. In the abstract as well as the text.

Response: The pre-specified non-inferiority margin (7%) was based on evidence.

The previous studies show the children who received CSB had a recovery rate from MAM of 67% and the recovery rate from MAM without any treatment was 54%, hence the difference is 13%. We divided 13% by two and got the number approaching 7%. Thus the non-inferiority margin was specified considering that the LIBS group could have at least a 7% higher recovery rate than in a group with no supplementary food (placebo). This was based on the recommendations of selecting a non-inferiority margin. This text was added to the revised manuscript. The revision is found in the manuscript on page 10, lines 196 to 202. The references we used are listed at the end of this paper and added in the revised manuscript (3,4,5).

2. Line 99: Did the LIBS contain vitamin and mineral supplements similar to the CBS+?

Response: There is no fortification of vitamins and minerals with LIBS. We determined the important nutrients (macronutrients and some minerals) found in the LIBS comparing with CSB+. In the study area, there is routine vitamin A supplementation for all children below 5 years of age. The nutrient compositions of both supplements were summarized in Table 1 on page 14, line 282.

 3. Line 111: replace demonstrate with determine.

Response:We rephrase it as to determine. The revision is found on page 6, line 96.

4. Line 250: the text read: this supplement..... does that mean one serving contained the mentioned supplements or was this daily contribution to total consumption per day? Please elaborate on this.

Response: It is only one serving for children but not a daily contribution to total consumption per day. We distributed the supplement in the morning to feed the children as a breakfast. The revision is found on page 13 and 14, lines 269, 272 to 273. 

Line 402: Provide the recovery rates of "MAM with child-centred counselling interventions only" to give a clear picture of the impact of the supplementary feeding.

Response: The recovery rates of "MAM with child-centered counseling intervention was 57.8%. This information is added to the text on page 22, line 443.

General comments:

1. Was there any part of the study that focused on relapse?

Response: We have no part of the study focused on relapse. 

2. Did the fieldworkers actually observe the consumption of the supplements at all times and of all meals? What was the possibility of sharing amongst other family members? I understand that provision was made for twins, but what about other family members?

Response: . The food distributors observe the consumption of supplements on daily basis. When there is leftovers, they facilitate to keep the food and re-feed the children. The information is found on page 14, lines 274 to 277, 281. To minimize sharing, we have distributed the supplement on daily basis, done daily follow-up during feeding, and added the amount of supplement if there are twins and other children with MAM. The information is found in the revised manuscript on page 10, lines 185 to 187; 11, lines 205 to 212 and page 9 and 0, lines 177 to 183.

3. Was the actual amount consumed measured or only the amount provided to the child? In other words, did the fieldworkers weigh food before and after consumption? The lack of additional food consumption data was a serious limitation so it is not clear if the children consumed only the supplement or additional family food.

Response: The food distributors measure the amount of consumed supplement using the local measuring cup. In case of not consuming all the prepared porridge, re-feeding was done. In the other hours of the day, the children ate their home diet. The participants from both groups were served the supplements in the morning as breakfast. This was mentioned on page 13, lines 269; 272 to 273. Not monitoring the children’s overall dietary consumption during the treatment and not having a non-supplemented control group was mentioned as the main limitation of the study on page 25, lines 502 to 504. 

 4. The randomisation of the included children was described in detail, but it is unclear how the Kebeles were selected. The paper (line 134) only states “selected kebeles” and line 189: randomly selected kebeles. Please elaborate on this as it might influence the representativeness of the study participants.

Response: Damot Pulassa district (woreda) was purposely selected based on a consideration of the high level of food insecurity, high level of child malnutrition, and access to transportation. All the kebeles in the district had a similar admission loads to the management program of MAM, for practical reasons, six out of the 23 kebeles were randomly selected for the study. The revision is found on page 8, lines 143 to 148.

5. It is unclear who knew the allocation of the two groups since the food distributors were also blinded. If the producers knew the difference between the two, when was allocation revealed? It is also not clear how many portion participants consumed per day (all at one meal, or spread throughout the day?)

Response: The allocation of two groups was done by one of the research team, who was not otherwise involved with the study. Mothers selected an envelope containing coded numbers consistent with one of the two supplementary foods. The person who did allocation was organized the food distribution process since he was aware of which number corresponded to which food but did not take part in the food distribution. This information is found on pages 11 to 12, lines 220 to 229. All children consume the supplement at one meal. When there were leftovers, re-feeding was done. This was mentioned on pages 13 to 14, lines 269, 272 to 273, and 280 to 281.

References

1. Dipasquale V, Cucinotta U, Romano C. Acute Malnutrition in Children: Pathophysiology, Clinical Effects and Treatment. Nutrients. 2020; 12(2413):1-9.

2. Georgief MK. Nutrition and the developing brain: nutrient priorities and measurement. Am J Clin Nutr. 2007; 85(suppl): 614S-20S.

3. Karakochuk C, van den Briel T, Stephens D, Zlotkin S. Treatment of moderate acute malnutrition with ready-to-use supplementary food results in higher overall recovery rates compared with a corn-soya blend in children in southern Ethiopia: an operations research trial. Am J Clin Nutr. 2012; 96:911–6.

4. James P, Sadlaw K, Wondafrash M, Argaw A, Luo H, Geleta B, Kedir K, Getnet Y, Belachew T, Bahwere P. Children with moderate acute malnutrition with no access to supplementary feeding programmes experience high rates of deterioration and no improvement: results from a prospective cohort study in rural Ethiopia. PLoS One. 2016;10:1371: 1-26.

5. Piaggio, G., Elbourne, DR, Pocock, SJ, Evans, SJW, Altman, DG, & The CONSORT Group. Reporting of noninferiority and equivalence: Randomized trials extension of the CONSORT 2010 statement. JAMA: Journal of the American Medical Association. 2010; 308(24): 2594–2604.

6. R.B. D’Agostino Sr., J.M. Massaro, L.M. Sullivan. Non-inferiority trials: design concepts and issues - the encounters of academic consultants in statistics. Stat Med, 22 (2003), pp. 169-186

---

## [Decision Letter · Decision Letter 1]

19 Apr 2021

PONE-D-20-29119R1

A local-ingredients-based supplement is an alternative to corn-soy blends plus for treating moderate acute malnutrition among children aged 6 to 59 months: A randomized controlled non-inferiority trial in Wolaita, Southern Ethiopia

PLOS ONE

Dear Dr. Nane,

Thank you for submitting your manuscript to PLOS ONE. After careful consideration, we feel that it has merit but does not fully meet PLOS ONE’s publication criteria as it currently stands. Therefore, we invite you to submit a revised version of the manuscript that addresses the points raised during the review process. Please make sure to address each of the comments carefully.

We look forward to receiving your revised manuscript.

Kind regards,

Seth Adu-Afarwuah

Academic Editor

PLOS ONE

Journal Requirements:

Reviewers' comments:

Reviewer's Responses to Questions

**Comments to the Author**

1. If the authors have adequately addressed your comments raised in a previous round of review and you feel that this manuscript is now acceptable for publication, you may indicate that here to bypass the “Comments to the Author” section, enter your conflict of interest statement in the “Confidential to Editor” section, and submit your "Accept" recommendation.

Reviewer #1: (No Response)

Reviewer #2: (No Response)

Reviewer #3: All comments have been addressed

Reviewer #4: All comments have been addressed

2. Is the manuscript technically sound, and do the data support the conclusions?

Reviewer #1: Yes

Reviewer #2: Yes

Reviewer #3: Yes

Reviewer #4: Yes

3. Has the statistical analysis been performed appropriately and rigorously? 

Reviewer #1: (No Response)

Reviewer #2: Yes

Reviewer #3: Yes

Reviewer #4: Yes

4. Have the authors made all data underlying the findings in their manuscript fully available?

Reviewer #1: Yes

Reviewer #2: Yes

Reviewer #3: Yes

Reviewer #4: Yes

5. Is the manuscript presented in an intelligible fashion and written in standard English?

Reviewer #1: Yes

Reviewer #2: No

Reviewer #3: Yes

Reviewer #4: Yes

6. Review Comments to the Author

Reviewer #1: For Table 2, statistical testing to be avoided for baseline comparison between the two groups. Baseline statistical testing is only applicable to cluster randomization trial. Please refer to CONSORT statement.

Reviewer #2: the format of the author response is brutal and extremely difficult to read. I find it very difficult to know what has been added into the revised version, because the additions are not indicated.

In my view, the revisions I requested are necessary to improve the organization, readability and presentation of the material. Merely adding a few words, or referring to another part of the manuscript is not OK.

If the authors are not willing to use the recommendations constructively, then the paper should be rejected.

Reviewer #3: (No Response)

Reviewer #4: Well done on the changes, it made a significant difference to the quality of the paper. Please have a look for a few minor editing problems.

7. PLOS authors have the option to publish the peer review history of their article (what does this mean?). If published, this will include your full peer review and any attached files.

Reviewer #1: No

Reviewer #2: **Yes: **Michael Lindinger

Reviewer #3: **Yes: **Getahun Ersino

Reviewer #4: **Yes: **Martani Lombard

---

## [Author Response · Author response to Decision Letter 1]

19 May 2021

PLOS ONE

19.05.21

Dear Editor 

Thank you for allowing us to submit a revised manuscript titled ‘A local-ingredients-based supplement is an alternative to corn-soy blends plus for treating moderate acute malnutrition among children aged 6 to 59 months: A randomized controlled non-inferiority trial in Wolaita, Southern Ethiopia’ to PLOS ONE journal. 

We are grateful to the editor and reviewers for their comments which allowed us to revise the manuscript. We have tried to improve the manuscript according to the comments raised by the reviewers. 

For the reference of pages and line numbers, we used a track changed version of the manuscript. In this letter, the reviewers’ comments are written in bold format and our point-by-point responses are written in normal font. The change indicated in the manuscript is included in this letter and written in italics. 

Yours sincerely,

Debritu Nane

Corresponding Author

 

Review Comments to the Author

Reviewer #1

For Table 2, statistical testing to be avoided for baseline comparison between the two groups. Baseline statistical testing is only applicable to cluster randomization trials. Please refer to the CONSORT statement.

Response: Statistical testing for baseline comparison between the LIBS and CSB+ groups was removed from the text and Table 2. The revision is found on page 18, line 381, page19, lines 390 to 391 (Table 2).

Reviewer #2 

The format of the author's response is brutal and extremely difficult to read. I find it very difficult to know what has been added to the revised version because the additions are not indicated. In my view, the revisions I requested are necessary to improve the organization, readability and presentation of the material. Merely adding a few words, or referring to another part of the manuscript is not OK. If the authors are not willing to use the recommendations constructively, then the paper should be rejected.

Response: Thank you for your constructive comments and recommendations that allow us to revise our manuscript and apologies for the inconveniences while reading the previous response letter. We have tried to revise our manuscript and respond in a readable way. Responses for the comments are specified/tracked in the manuscript.

Page 2 line 22 (in reviewed version): Define wasted scientifically, and/or replace in the abstract with something that malnourished

Response: We have now used the terminology acute malnutrition throughout the paper. However, acute malnutrition means the same as wasting. This revisions are now found on page 4, lines 54, 55, 61, 65, and 71 (in revised version with track changes).

Page 2 line 33 (in reviewed version): Is the sugar added to LIBS sucrose! Toxic

Response: We have added cane sugar to the LIBS for two reasons. The first one is to improve the taste of LIBS. The second one is to improve or meet the calorie requirement that should come from carbohydrates, and to make it comparable with the CSB+. CSB+ is the conventional food provided for children with MAM in the control group. Regarding toxicity, the level of carbohydrates found in the LIBS is still lower than in CSB+.

This revision now reads as “The cane sugar was added to the LIBS in which the taste of LIBS made better and the amount of calorie that should come from carbohydrate was improved but still lower than the level of carbohydrate found in the CSB+ (conventional food provided for children with MAM in the control group.” This clarification is found in the revised manuscript with track changes on pages 14 to 15, lines 295 to 298 and page 15, line 309. Table 1

Page 2 line 33 (in reviewed version): Specify oil

Response: We included the revised sentence in the abstract part. We used refined deodorized and cholesterol-free sunflower oil. 

The revised manuscript now reads as “In the ﬁrst arm, 125.2 g of LIBS with 8 ml of refined deodorized and cholesterol-free sunflower oil/day was provided. In the second arm, 150 g of CSB+ with 16 ml of refined deodorized and cholesterol-free sunflower oil/day was provided”. This revision is found on page 2, lines 32 to 35 (in revised version with track changes). 

In the method section, we described the constituents of each intervention and specified what sugar and oil are. 

The change made in the manuscript is as follows:

“Subjects in the intervention group received a daily ration of 125.2 g of LIBS with 8 ml of refined deodorized and cholesterol free sunflower oil for 12 weeks. The composition of LIBS was: 30 g of pumpkin seed, 25 g of peanut grain, 20 g of amaranth grain, 15 g of flaxseed, 10 g of emmer wheat, and 25.2 g of cane sugar with 8 ml of refined deodorized and cholesterol free sunflower oil. This supplement (one serving) yielded 699 kcal, 22.6 g protein, 56.9 g carbohydrate, and 40.89 g fat. The cane sugar was added to the LIBS in which the taste of LIBS made better and the amount of calorie that should come from carbohydrate was improved but still lower than the level of carbohydrate found in the CSB+ (conventional food provided for children with MAM in the control group). Likewise, children in the control group received 150 g of CSB+/day with 16 ml of refined deodorized and cholesterol-free sunflower oil; this yielded 751 kcal, 21.25 g protein, 95 g carbohydrate, and 31.76 g fat daily for 12 weeks”. This revised paragraph is found on pages 14 and 15, lines 290 to 301 (in revised version with track changes).

Page 4 line 55 (in reviewed version): Explain what aspect of impaired growth and development is “acute”. Malnutrition must be chronic and or frequently repeated for impaired growth and development.

Response: During acute malnutrition, there could be childhood infections that lead to death, debility and impairment in growth and development. When a repeated episode of acute malnutrition occurs, often in combination with infections, it can lead to stunting (chronic malnutrition). Sometimes, especially if the child is less than two years of age, it can also affected other developmental features such as brain impairment. 

This information is added to the revised text with track changes on page 4, lines 61 to 64. It now reads as “When a repeated episode of acute malnutrition occurs, often in combination with infections, it can lead to stunting (chronic malnutrition). Sometimes, especially if the child is less than two years of age, it can also affect other developmental features such as brain impairment”. 

Page 4 line 63 (in reviewed version): Acute means what in terms of nutrient delivery?

Response: Acute malnutrition is a recent thinness caused by inadequate intake of macronutrients (energy and protein), infections and other several factors. This information was added to the revised manuscript page 4, lines 55 to 58 (in revised version with track changes). 

The information is as follows:

“AM often occurs when there is severe weight loss possibly caused by inadequate energy and protein intake, and infections. Such events frequently occur during periods of prolonged food insecurity, poverty, poor feeding practices, and inadequate household food accessibility”. 

Page 5 lines 86, 88 and 89 (in reviewed version): Specify oil and sugar found in CSB+ and CSB++ 

Response: The specification of oil and sugar has now been added to the conventional food supplements (CSB+ and CSB++) for children with MAM were not mentioned in the literatures. Different literatures described the constituents of CSB, CSB+, and CSB++ but they did not specify which kind of sugars and oils that were used. The paragraph with the references is found in the manuscript, introduction part, page 5 and 6, lines 94 to 100 (in revised version with track changes). In our study, children with MAM got refined deodorized and cholesterol free sunflower oil during CSB+ ration but they may not be provided with sugar. 

Page 5 line 91 (in reviewed version): What cereal and grains that CSB+ made up of

Response: CSB+ is a combination of the cereal (maize or corn) and a legume, most often soybean. The maize or corn found in CSB+ could be named as cereals and grains interchangeably. 

The revised text now reads as “The combinations of the cereal (corn or maize) and the legume (soybean) was refined, blended, and precooked (by roasting) for children with MAM below five years of age”. This revision is found on page 6, lines 103 to 105 (in revised version with track changes).

Page 7 line 120 (in reviewed version): Introduction needs better focus; reduce length by at least 30%; restrict to the introduction, not the discussion

Response: To reduce the length of the introduction, we have removed some information from the text and moved into conclusion part. The revisions were indicated in the track changed manuscript of page 4, lines 58 to 61; pages 4,lines 71 to 73; page 5, lines 75 to 83 and lines 91 to 96; pages 6 and 7, lines 116 to 122, and page 7, lines 129 to 131 (in revised version with track changes). 

Some information has also been moved to the conclusion part. 

The revised and moved text is as follows:

“The evidence drawn from this study will be shared with the public and with policymakers as LIBS has the potential to increase recovery from MAM and decrease the burden of malnutrition”. This revision is found on page 27, lines 546 to 548 (in revised version with track changes).

Page 7 line 121 (in reviewed version): Requirement of ethical statement at the beginning of methods part

Response: The ethics approval and consent to participate statement is moved to the first part of the method section. 

The paragraph in the revised manuscript is as follows:

“This study was approved by the Hawassa University College of Medicine and Health Sciences Institutional Review Board (IRB/024/10) and Regional Committees for Medical and Health Research Ethics in Norway (2018/69/REK vest).The approval obtained from the institutional review boards covered all sites included in the study. The purpose of the study and methods of data collection, confidentiality, and voluntary participation were explained to the mothers of children who were invited to sign an informed consent form. Verbal (getting thumb mark after reading the information) and written informed consent were obtained from all caregivers of children who met enrollment criteria before the recruitment of their children into the study. All interviews and intervention procedures were conducted in privacy. This trial was registered at Pan-African Clinical Trial Registration as PACTR201809662822990. The authors confirmed that all ongoing and related trials for these food supplements were registered”. This revision is found on page 8, lines 140 to 149 (in revised version with track changes).

Page 12 line 249 (in reviewed version): Provide details- specifics about the sugar and oil

Response: We accepted the comment and specified the oil and sugar given to the subjects in the abstract and method parts. We used refined deodorized and cholesterol-free sunflower oil for both groups. Regarding sugar, we used refined cane sugar. 

The revisions are as follows:

“In the ﬁrst arm, 125.2 g of LIBS with 8 ml of refined deodorized and cholesterol-free sunflower oil/day was provided. In the second arm, 150 g of CSB+ with 16 ml of refined deodorized and cholesterol-free sunflower oil/day was provided”. This revision is found in the abstract part page 2, lines 32 to 36 (in revised version with track changes).

“This was a randomized, controlled, non-inferiority trial that assessed the efficacy of 125.2 g of LIBS with 8 ml of refined deodorized and cholesterol-free sunflower oil/day (the intervention), compared with conventional treatment, which is CSB+ in the amount of 150 g of CSB+/day with 16 ml of refined deodorized and cholesterol-free sunflower oil (the control), in treating MAM for 12 weeks”. This revision is found in the method part under the study design on pages 9 and 10, lines 181 to 185 (in revised version with track changes).

In the method section under the description of the intervention, we have revised a paragraph. 

Now it reads as “Subjects in the intervention group received a daily ration of 125.2 g of LIBS with 8 ml of refined deodorized and cholesterol free sunflower oil for 12 weeks. The composition of LIBS was: 30 g of pumpkin seed, 25 g of peanut grain, 20 g of amaranth grain, 15 g of flaxseed, 10 g of emmer wheat, and 25.2 g of cane sugar with 8 ml of refined deodorized and cholesterol free sunflower oil. This supplement (one serving) yielded 699 kcal, 22.6 g protein, 56.9 g carbohydrate, and 40.89 g fat. The cane sugar was added to the LIBS in which the taste of LIBS made better and the amount of calorie that should come from carbohydrate was improved but still lower than the level of carbohydrate found in the CSB+ (conventional food provided for children with MAM in the control group). Likewise, children in the control group received 150 g of CSB+/day with 16 ml of refined deodorized and cholesterol-free sunflower oil; this yielded 751 kcal, 21.25 g protein, 95 g carbohydrate, and 31.76 g fat daily for 12 weeks.” This revision is found on pages 14 to 15, lines 290 to 301 (in revised version with track changes).

Page 12 line 251 (in reviewed version): Specify what flour is

Response: Previously the sentence was written as follows: Children in the control group received CSB+ in the amount of 150 g flour/day with 16 ml of oil. We removed the term flour since it is unnecessary word. 

Now it is revised as “children in the control group received 150 g of CSB+/day with 16 ml of refined deodorized and cholesterol-free sunflower oil; this yielded 751 kcal, 21.25 g protein, 95 g carbohydrate, and 31.76 g fat daily for 12 weeks. This revision is found on page 15, lines 298 to 301 (in revised version with track changes).

Page 12 lines 250 and 252 (in reviewed version): The amount of carbs and ash in LIBS and CSB+

Response: the amount of carbohydrate and ash found in both supplements is now mentioned in the revised manuscript. The amount of carbohydrate found in the LIBS is 56.9 g whereas the amount of carbohydrate found in the CSB+ is 95 g. The ash content in the CSB+ is higher (4.3g) than LIBS (2.1g). 

These revisions are found on page 15, line 311 (in revised version with track changes). (Table 1)

Here is the table.

Table 1. Nutrient composition of the supplementary foods.

Nutrient 125 gm of LIBS with 8 ml oil 150 gm of CSB+ with 16ml of oil

Energy (kcl) 698.5 750.84

Protein (g) 22.6 21.25

Carbohydrate (g) 56.9 95.0

Fat (g) 40.89 3176

Ash (g) 2.1 4.3

Iron (mg) 8.1 6.00

Zn (mg) 5.6 7.5

Calcium (mg) 100.00 195.00

Phosphorous (mg) 470.55 300.00

Potassium (mg) 666.14 600.00

Magnesium (mg) 394.7 107.75 

Sodium (mg) 84.6 41.25

Folic acid (µg) 49.4 90.00

Major recommendations:

1. do a thorough job of defining MAM - it is incomplete

Response: Previously, the definition of MAM was written in the text as “Moderate acute malnutrition (MAM) as weight-for-height z-score (WHZ) between -3 and -2 standard deviations and/or mid-upper arm circumference (MUAC) of between ≥11.5 cm and <12.5 cm”. To make the definition of MAM complete and differentiate it from severe form, we have added the information about edema. 

The revised definition is as follows:

“Moderate acute malnutrition (MAM) is defined as weight-for-height z-score (WHZ) between -3 and -2 z-scores of the WHO Child Growth Standards median and/or mid-upper arm circumference (MUAC) of between greater or equal to 11.5 cm and less than 12.5 cm, without bipedal edema”. The reference is added to the manuscript. (This information is found on page 4, lines 65 to 68 in revised version with track changes).)

2. do a thorough job of specifying what is meant by sugar, oil, vegetable oil - these are general terms. 

Response: The oil that we used for both groups was refined deodorized and cholesterol-free sunflower oil. The sugar that we used was cane sugar. We used cane sugar to improve the taste of LIBS and to make the amount of calories that should come from carbohydrates comparable with the CSB+ (conventional food provided for children with MAM in the control group). 

The revisions that are found in the manuscript are as follows:

“In the ﬁrst arm, 125.2 g of LIBS with 8 ml of refined deodorized and cholesterol-free sunflower oil/day was provided. In the second arm, 150 g of CSB+ with 16 ml of refined deodorized and cholesterol-free sunflower oil/day was provided.” (This information is found on page 2, lines 32 to 35 in revised version with track changes)

“This was a randomized, controlled, non-inferiority trial that assessed the efficacy of 125.2 g of LIBS with 8 ml of refined deodorized and cholesterol-free sunflower oil/day (the intervention), compared with conventional treatment, which is CSB+ in the amount of 150 g of CSB+/day with 16 ml of refined deodorized and cholesterol-free sunflower oil (the control), in treating MAM for 12 weeks”. (This information is found on page 9 to 10, lines 181 to 185 in revised version with track changes)

“Subjects in the intervention group received a daily ration of 125.2 g of LIBS with 8 ml of refined deodorized and cholesterol free sunflower oil for 12 weeks. The composition of LIBS was: 30 g of pumpkin seed, 25 g of peanut grain, 20 g of amaranth grain, 15 g of flaxseed, 10 g of emmer wheat, and 25.2 g of cane sugar with 8 ml of refined deodorized and cholesterol free sunflower oil. This supplement (one serving) yielded 699 kcal, 22.6 g protein, 56.9 g carbohydrate, and 40.89 g fat. The cane sugar was added to the LIBS in which the taste of LIBS made better and the amount of calorie that should come from carbohydrate was improved but still lower than the level of carbohydrate found in the CSB+ (conventional food provided for children with MAM in the control group). Likewise, children in the control group received 150 g of CSB+/day with 16 ml of refined deodorized and cholesterol-free sunflower oil; this yielded 751 kcal, 21.25 g protein, 95 g carbohydrate, and 31.76 g fat daily for 12 weeks.” (Pages 14 to 15, lines 290 to 301 in revised version with track changes)

3. focus the Introduction - some of the info here is better combined and condensed into the Discussion

Response: 

To reduce the length of the introduction, we have removed some information from the text and moved into conclusion part. The revisions were indicated in the track changed manuscript of page 4, lines 58 to 61; pages 4,lines 71 to 73; page 5, lines 75 to 83 and lines 91 to 96; pages 6 and 7, lines 116 to 122, and page 7, lines 129 to 131 (in revised version with track changes). 

Some information has also been moved to the conclusion part. 

The revised and moved text is as follows:

“The evidence drawn from this study will be shared with the public and with policymakers as LIBS has the potential to increase recovery from MAM and decrease the burden of malnutrition”. This revision is found on page 27, lines 546 to 548 (in revised version with track changes).

Reviewer #3: (No Response)

Reviewer #4: Well done on the changes, it made a significant difference to the quality of the paper. Please have a look for a few minor editing problems.

---

## [Decision Letter · Decision Letter 2]

21 Jul 2021

PONE-D-20-29119R2

A local-ingredients-based supplement is an alternative to corn-soy blends plus for treating moderate acute malnutrition among children aged 6 to 59 months: A randomized controlled non-inferiority trial in Wolaita, Southern Ethiopia

PLOS ONE

Dear Dr. Nane,

Thank you for submitting your manuscript to PLOS ONE. After careful consideration, we feel that it has merit but does not fully meet PLOS ONE’s publication criteria as it currently stands. Therefore, we invite you to submit a revised version of the manuscript that addresses the points raised during the review process.

We look forward to receiving your revised manuscript.

Kind regards,

Miquel Vall-llosera Camps

Senior Editor

PLOS ONE

Additional Editor Comments:

Please address the language/syntax issues raised by Reviewer#3.

Journal Requirements:

Reviewers' comments:

Reviewer's Responses to Questions

**Comments to the Author**

1. If the authors have adequately addressed your comments raised in a previous round of review and you feel that this manuscript is now acceptable for publication, you may indicate that here to bypass the “Comments to the Author” section, enter your conflict of interest statement in the “Confidential to Editor” section, and submit your "Accept" recommendation.

Reviewer #1: All comments have been addressed

Reviewer #2: All comments have been addressed

Reviewer #3: All comments have been addressed

2. Is the manuscript technically sound, and do the data support the conclusions?

Reviewer #1: (No Response)

Reviewer #2: Yes

Reviewer #3: Yes

3. Has the statistical analysis been performed appropriately and rigorously? 

Reviewer #1: (No Response)

Reviewer #2: Yes

Reviewer #3: Yes

4. Have the authors made all data underlying the findings in their manuscript fully available?

Reviewer #1: (No Response)

Reviewer #2: Yes

Reviewer #3: Yes

5. Is the manuscript presented in an intelligible fashion and written in standard English?

Reviewer #1: (No Response)

Reviewer #2: Yes

Reviewer #3: Yes

6. Review Comments to the Author

Reviewer #1: (No Response)

Reviewer #2: Thank you very much for an excellent revision yyyyyyyyyyyyyyyyyyyyyyyyyyyyyyyyyyyyyyyyyyyyyyyyyyyyyyy .

Reviewer #3: Authors have addressed comments and concerns raised in the initial review. However a thorough editing of the paper for language/syntax would increases its readability. Please avoid using some unusual/uncommon abbreviations (e.g., AM to refer to acute malnutrition. I don't see any value in abbreviating it other than adding confusion). Use abbreviation only where needed. I trust the authors would address these comments in their own terms before the paper gets published and I do not expect a separate response here.

7. PLOS authors have the option to publish the peer review history of their article (what does this mean?). If published, this will include your full peer review and any attached files.

Reviewer #1: No

Reviewer #2: **Yes: **Michael I Lindinger

Reviewer #3: **Yes: **Getahun Ersino

---

## [Author Response · Author response to Decision Letter 2]

19 Aug 2021

Thank you for letting us to submit the revised version of our manuscript. The reviewer (reviewer 3) who requested the revisions of grammatical issues has mentioned that he does not expect a separate response. That is why we didn't submit the separate response letter but we have submitted the revised manuscript that indicate the changes made on the grammatical issues.

---

## [Editor Report · Decision Letter 3]

5 Oct 2021

A local-ingredients-based supplement is an alternative to corn-soy blends plus for treating moderate acute malnutrition among children aged 6 to 59 months: A randomized controlled non-inferiority trial in Wolaita, Southern Ethiopia

PONE-D-20-29119R3

Dear Dr. Nane,

We’re pleased to inform you that your manuscript has been judged scientifically suitable for publication and will be formally accepted for publication once it meets all outstanding technical requirements.

Kind regards,

Miquel Vall-llosera Camps

Senior Editor

PLOS ONE
---

## [Editor Report · Acceptance letter]

19 Oct 2021

PONE-D-20-29119R3 

A local-ingredients-based supplement is an alternative to corn-soy blends plus for treating moderate acute malnutrition among children aged 6 to 59 months: A randomized controlled non-inferiority trial in Wolaita, Southern Ethiopia 

Dear Dr. Nane:

I'm pleased to inform you that your manuscript has been deemed suitable for publication in PLOS ONE. Congratulations! Your manuscript is now with our production department. 

Kind regards, 

on behalf of

Dr. Miquel Vall-llosera Camps 

Staff Editor

PLOS ONE